# Alcohol use and associated risk factors among female sex workers in low- and middle-income countries: A systematic review and meta-analysis

**Alicja Beksinska** [ID]*, **Oda Karlsen, Mitzy Gafos, Tara S. Beattie** [ID]

Department of Global Health and Development, Faculty of Public Health and Policy, London School of Hygiene and Tropical Medicine, London, United Kingdom

* Alicja.Beksinska@nhs.net

**Data Availability Statement:** This is a systematic review with all data used extracted from existing studies. The data underlying the quantitative

## Abstract

Due to its widespread use in the sex work industry, female sex workers (FSWs) in low- and middle-income countries (LMICs) are at high risk of harmful alcohol use and associated adverse health outcomes. Factors associated with harmful alcohol use include violence, mental health problems, drug use, sexual risk behaviors and HIV/STIs. To our knowledge, there has been no quantitative synthesis of FSW alcohol use data to date. This systematic review and meta-analysis aims to provide an estimate of the prevalence of harmful alcohol use among FSWs in LMICs and to examine associations with common health and social concerns. The review protocol was registered with PROSPERO, number CRD42021237438. We searched three electronic databases for peer-reviewed, quantitative studies from inception to 24th February 2021. Studies were selected for inclusion that reported any measure of prevalence or incidence of alcohol use among FSWs aged 18 or older from countries defined as LMIC in accordance with the World Bank income groups 2019. The following study designs were included: cross-sectional survey, case–control study, cohort study, case series analysis, or experimental study with baseline measures for alcohol use. Study quality was assessed with the Center for Evidence-Based Management (CEBMa) Critical Appraisal Tool. Pooled prevalence estimates were calculated for (i) any hazardous/harmful/dependent alcohol use, (ii) harmful/dependent alcohol use only, both overall and by region and (iii) daily alcohol use. Meta-analyses examined associations between harmful alcohol use and violence, condom use, HIV/STIs, mental health problems and other drug use. In total, 435 papers were identified. After screening, 99 papers reporting on 87 unique studies with 51,904 participants from 32 LMICs met the inclusion criteria. Study designs included cross-sectional (n = 89), cohort (n = 6) and experimental (n = 4). Overall, 5 scored as high quality, 79 studies scored as moderate and 15 scored as weak quality. Twenty-nine papers reporting on 22 unique studies used validated alcohol use tools including AUDIT, CAGE and WHO CIDI. The pooled prevalence of any hazardous/harmful/dependent alcohol use was 41% (95% CI: 31–51%), and of daily alcohol use was 26% (95% CI: 17–36%). There was variation in harmful alcohol use by global region (Sub-Saharan Africa: 38%; South Asia/Central Asia/ East Asia and Pacific: 47% and Latin America and the

synthesis are provided in Tables 1 and 2 within the manuscript.

**Funding:** Funding for this study was provided by the Medical Research Council and the UK Department of International Development (DFID) (MR/R023182/1) under the MRC/DFID Concordat agreement. No funding bodies had any role in study design, data collection and analysis, decision to publish, or preparation of the manuscript.

**Competing interests:** The authors have declared that no competing interests exist.

Caribbean:44%). Harmful alcohol use was significantly associated with inconsistent condom use (pooled unadjusted RR: 1.65; 95% CI: 1.01–2.67), STIs (pooled unadjusted OR: 1.29; 95% CI 1.15–1.46); and other drug use (pooled unadjusted OR of 2.44; 95% CI 1.24–4.80), but not with HIV, violence or mental health problems. We found a high prevalence of problem alcohol use and daily alcohol use among FSWs in LMICs. Harmful drinking was associated with important HIV risk factors such as inconsistent condom use, STIs and other drug use. Major limitations included heterogeneity in tools and cut-off scores to measure alcohol use and other common risk factors, and a paucity of longitudinal studies. There is an urgent need for tailored interventions for FSWs in LMICs that address alcohol use as well as the associated sex work risk environment.

## Introduction

Harmful alcohol use is a major public health concern globally, contributing to 3 million deaths every year [1] and increasing the risk of many non-communicable diseases such as liver disease, infectious diseases such as human immunodeficiency virus (HIV) [2], mental health problems such as depression, and harm from external causes such as injuries and violence [3]. Among people aged 15–49 it is the leading risk factor for premature mortality and disability [1, 4]. In recent years alcohol use has been increasing in many LMICs [3]. Increased alcohol regulations to combat alcohol-related harms in high income countries [3] has led the alcohol industry to seek new sources for profit amongst populations with previously lower levels of alcohol consumption [5, 6]. This has concerning implications for both direct and indirect alcohol-related harms [5, 7]. Alcohol use patterns and alcohol-related harms vary globally and by gender and socio-economic status. In a multi-country European study, lower educated men were more at risk of heavy episodic drinking whilst amongst women, higher education was associated with heavy drinking [8]. In sub-Saharan Africa modelling has shown that high-income earners have the highest prevalence of any alcohol use, while low-income earners consume more alcohol per person, and have a higher burden of alcohol-related harm [9]. The need to address alcohol use is included in the Sustainable Development Goals [10], highlighting increased awareness of the burden of alcohol and related harms in LMICs, particularly among high risk groups.

Sex work—defined as the receipt of money or goods in exchange for sexual services—is criminalised in most parts of the world [11, 12]. FSWs face unique occupational risks including sexual and physical violence from clients, and high levels of HIV and other sexually transmitted infections (STI) as well as structural inequalities including police arrest, discrimination, poverty, and gender inequality. In addition, alcohol is widely available in the sex work industry [13, 14] with sex work commonly taking place in venues such as bars with high alcohol availability and women reporting alcohol use to cope with the daily challenges of sex work [14–17]. These factors may predispose FSWs to increased risks of harmful alcohol use. Socio-cultural and economic factors have an effect on the structural and occupational risks associated with sex work [12, 16]; for example differing levels of sex work criminalisation and access to sexual health services mean that alcohol use and associated risks are likely to differ for sex workers in LMICs compared to those in high-income countries. In 2010, Li et al. conducted an integrative review exploring the use of alcohol among FSWs globally, and reported that 81.2–100% of FSWs had ever used alcohol and 73.3–74.8% had used alcohol in the past month [14]. However, the review highlighted several limitations in the currently available literature including

the lack of use of validated measurement tools and no meta-analysis was carried out. Additionally, the review did not disaggregate data by low income vs. high income settings. Many studies used very general terms for alcohol use with no further details in quantity, frequency or specified time period. Overall, they reported that problem drinking was under-investigated with no studies using validated tools such as the Alcohol Use Disorders Identification Test (AUDIT) tool [17] to quantify hazardous or harmful use.

Definitions of problem alcohol use vary in the literature, and encompass a spectrum from hazardous to harmful to dependent alcohol use that corresponds to the AUDIT tool (the most commonly used alcohol measurement tool), and other alcohol use tools such as CAGE (Cutting down, Annoyance, Guilty feelings and an Eye-opener) [18] and CIDI (Composite International Diagnostic Interview) [19]. Hazardous alcohol use (AUDIT $\geq 7/\geq 8$, AUDIT-C $\geq 3$, CAGE $\geq 2$) is usually defined as a pattern of alcohol consumption that increases someone's risk for physical and/or psychological harm. Harmful alcohol use (AUDIT score 16–19) is defined as a pattern of alcohol consumption that is causing mental or physical health problems, while alcohol dependence (AUDIT score > 20) is defined as craving, tolerance, and preoccupation with alcohol alongside continued drinking in spite of harmful consequences [20, 21]. Since Li's review there have been a range of studies among FSWs in LMICs reporting on alcohol use with an increase in the use of validated tools such as AUDIT. A recent systematic review of alcohol use among occupational groups at high risk of HIV in sub-Saharan Africa [22] reported that the pooled prevalence of alcohol misuse among sex workers was 45.3% (IQR 25.1–52.0%, 12 studies) but this was not specific to FSWs and was focussed only on sub-Saharan Africa. In addition to harmful alcohol use, there is substantial evidence demonstrating that FSWs in LMICs experience multiple stressors including poverty, lack of education, gender-based violence, mental health problems, drug use, high rates of HIV and STIs and stigma and discrimination [12, 16, 23, 24]. Many of the interconnected health and social issues faced by FSWs can be considered through a syndemics framework. Syndemics are defined by the clustering of two or more diseases in a population, the biological, social and psychological interaction between those diseases and the wider political and socio-economic context that drive the risk of these diseases [25, 26]. As a result, it is important to consider not just the burden of alcohol use, but the key associated risk factors. The review from Li et. al and several other studies among FSWs have reported common associations between alcohol use and poor physical health [14], illicit drug use [27, 28], mental health problems, violence [27], low condom use [27, 29–32], and HIV/STIs [27, 33] but there has been no synthesis of this evidence to date. The need for this systematic review was identified as part of the Maisha Fiti Study, a mixed-method, longitudinal study with FSWs in Nairobi examining associations between violence, mental health, harmful alcohol and other drug use, biological changes to the immune system and HIV/STI prevalence and risk factors. Recently published data from baseline, found that a third (29.9%; 95%CI 27.0–32.6%) of FSWs reported harmful (moderate/high risk) alcohol use, according to the WHO ASSIST tool [34]. Findings from Maisha Fiti identified a gap in the literature on the burden of alcohol use among FSWs, its associated risks and the need for evidence-based interventions.

High quality evidence on the burden of alcohol use among FSWs is key at the global and country level to guide policymaking and develop interventions tailored to address the syndemic health and social challenges faced by FSWs. This systematic review aims to provide an estimate of the prevalence of harmful alcohol use among FSWs in LMICs, and to examine associations with common health and social concerns among this group such as violence, condom use and HIV/STIs to inform intervention development.

## Methods

### Search strategy and selection criteria

The review protocol is registered with PROSPERO, number CRD42021237438 (https://www.crd.york.ac.uk/prospero/). We used the Preferred Reporting Items for Systematic reviews and Meta-Analysis (PRISMA) guidelines (Fig 1). We searched three electronic peer-reviewed literature databases include Ovid (EMBASE, PsycINFO, Global Health, Ovid MEDLINE), PubMed and Web of Science from first record until 24.02.2021. The following search terms were used: AUD OR "alcohol use disorder*" OR alcoholism OR alcohol addict* OR alcohol abuse OR alcohol dependence OR alcohol misuse OR heavy drinking OR binge drink*; "sex work*" OR "female sex work*" OR "FSW*" OR "prostitut*" OR "female prostitut*" OR "sex trad*" OR "transact* sex" OR "commercial sex" OR "sex-trade worker*"; "developing countr*" OR "less developed countr*" OR "under developed countr*" OR "underserved countr*" OR "deprived countr*" OR "poor countr*" OR "transition* countr*" OR "names of countries which fit world bank criteria for LMIC". (see S1 Appendix for search strategy). We additionally utilised citation list searching to source other eligible studies, as per PRISMA guidelines [35].

### Inclusion criteria

- This review included studies that reported any measure of prevalence or incidence of alcohol use or associations with alcohol use on the basis of a clinical interview, self-reported or clinical examinations among FSWs even if sex workers were not the main focus of the study.

- Studies were included from countries defined as low or middle income, in accordance with the World Bank income groups 2019 [36].

- Eligible studies had to be peer-reviewed.

- Eligible studies included females aged 18 or older who were actively engaged in sex work.

- The following study designs were included: cross-sectional survey, case–control study, cohort study, case series analysis, or experimental study with baseline measures for alcohol use.

- Studies were limited to English language.

### Exclusion criteria

- FSWs who identified as trans sex workers were excluded. This is because transgender sex workers risks and experiences of sex work are considered to be significantly different.

- We excluded studies that used qualitative methods only, were review papers, conference abstracts or non-peer reviewed publications.

- Studies not disaggregating data by alcohol for example referring to 'alcohol/drug use' were excluded from this review.

- Studies focused on women engaged in transactional sex only, were ineligible for review, as this practice, and its implications on health, is distinct from sex work [37].

Two reviewers (OK and AB) independently screened all publications in Covidence (https://www.covidence.org/reviewers/) according to the inclusion/exclusion criteria. If Covidence reported a conflict between reviewers on whether a study should be included, the abstract was reviewed, discussed and a final decision reached. Once abstracts were screened, the authors reviewed the full text for final eligibility check.

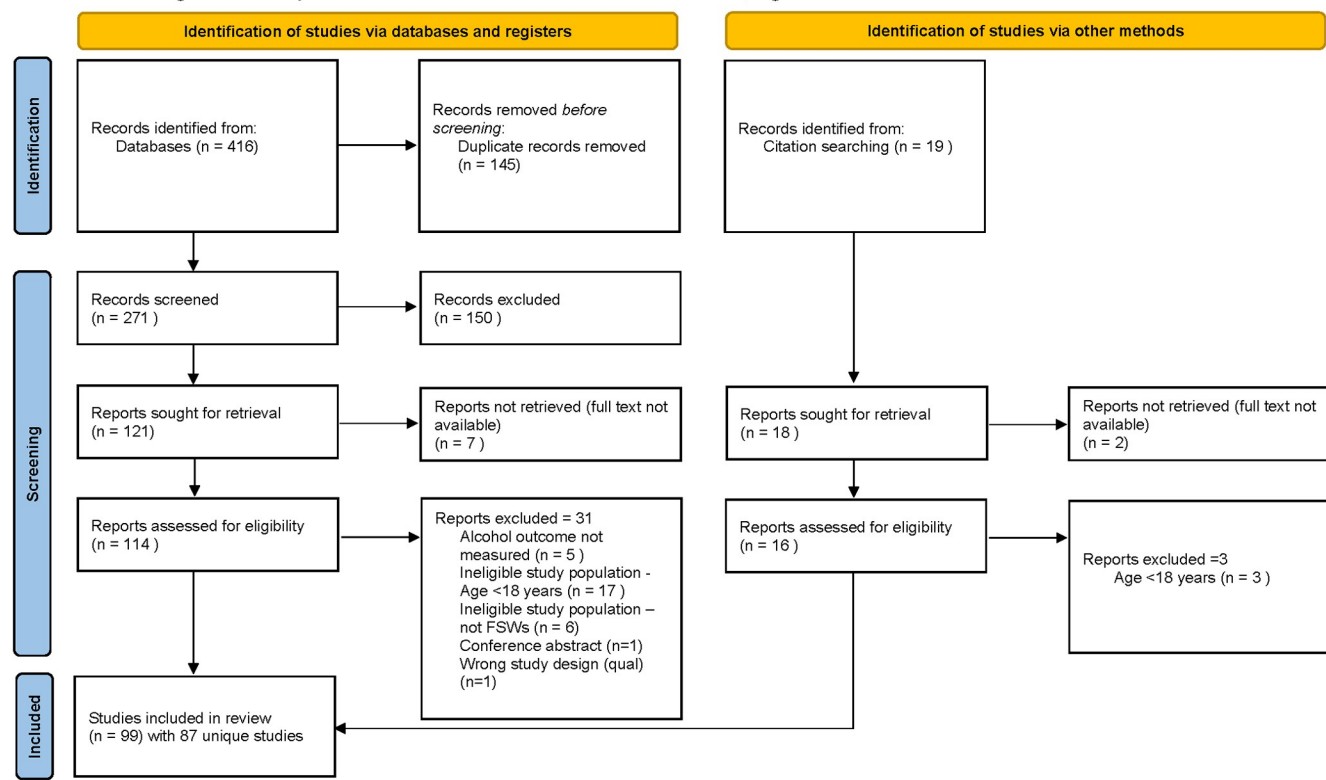

**Fig 1. Study selection.** PRISMA 2020 flow diagram.

## Quality assessment

Study quality was assessed by two authors (OK and AB) using the Center for Evidence-Based Management (CEBMa) Critical Appraisal for Cross-Sectional Surveys Tool. Assessment criteria included questions on study design, selection bias, statistical power, validity and reliability of measurement tools, statistical significance and confounding (see S2 Appendix for full details of CEBMa tool). The last item on CEBMa was removed (Item 12: "Can the results be applied to your organisation?") as it was not relevant to this review. OK and AB each scored half the studies–authors compared 10% of the results of scoring and discussed disagreements in scoring to ensure uniformity in the quality assessment process. Each study was rated based on 11 items, and an overall score was calculated. Studies scoring ≥8 out of 11 points were considered high quality, between 5 and 7 were rated moderate quality, and ≤4 were scored as weak quality. Scoring was based on cut offs used in a previous systematic review among FSWs [16]. A breakdown of individual scores and differences in author scoring are shown in Tables 1 and 2, S3 Appendix. Studies scoring as weak quality were not included in the meta-analyses.

## Data extraction and analysis

Data were extracted by two authors (OK and AB) into a structured data extraction document (Table 1) to include data on: author, study design, publication year, country of publication, study design, sample size, alcohol use measure and types, prevalence of alcohol use, and

**Table 1. Study characteristics.**

| Author, year & study design | Country | Sample | Sampling Procedure | Outcome(s) of interest | Events | Sample size | Event rate (%) | Method of assessing outcome (s) | Research quality |
|---|---|---|---|---|---|---|---|---|---|
| SUB-SAHARAN AFRICA | | | | | | | | | |
| Bazzi (2019) *Cross sectional* | Kenya | FSWs | Outreach and targeted sampling + snowball sampling and purposive sampling | Alcohol use in the past month | 45 | 45 | 100.0% | Semi-structured interviews | Low (3) |
| | | | | Always/often drunk when using alcohol | 25 | 45 | 56.0% | | |
| Bitty-Anderson (2019)[i] *Cross sectional* | Togo | FSWs | Time-location sampling | Moderate drinking | 275 | 937 | 29.4% | AUDIT score 1-6 | High (9) |
| | | | | Hazardous consumption | 344 | 937 | 36.7% | AUDIT score > 7 | |
| | | | | Binge drinking | 406 | 937 | 43.4% | Consumption of six or more alcohol drinks at least once per month in one occasion (AUDIT item nr 3). | |
| Tchankoni (2020)[i] *Cross sectional* | Togo | FSWs | Venue-based sampling | Moderate drinking | 197 | 952 | 20.7% | AUDIT-C score 1-3 | Moderate (7) |
| | | | | Hazardous consumption | 432 | 952 | 45.5% | AUDIT-C using cut off >4. | |
| Bukenya (2013) *Cross sectional* | Uganda | FSWs | Targeted and snowball sampling | Alcohol use | | | | Structured face-to-face interviews | Moderate (5) |
| | | | | *Less than once a week* | 60 | 905 | 6.6% | | |
| | | | | *At least once a week* | 421 | 905 | 46.5% | | |
| | | | | *Daily* | 235 | 905 | 26.0% | | |
| Bukenya (2019) *Cross sectional* | Uganda | FSWs | Convenience sampling | Hazardous alcohol use | 462 | 819 | 56.4% | AUDIT Score ≥ 7 | High (9) |
| Chersich (2007) *Cross sectional* | Kenya | FSWs | Snowball sampling | Drink but do not binge drink | 312 | 719 | 43.3% | Structured questionnaire Based on WHO definitions of alcohol • Never drunk alcohol • non-binge drinkers = lifetime use of alcohol but of <five drinks on any occasion in the preceding month • Binge drinkers (>five drinks on > one occasion in the previous month • previous month | Moderate (5) |
| | | | | Binge drink | 230 | 719 | 32% | | |
| | | | | Ever drunk alcohol | 542 | 719 | 75.4% | | |
| | | | | *Current drinking frequency;* | | | | | |
| | | | | Secondary abstinence | 82 | 542 | 15.1% | | |
| | | | | One to three times a month | 91 | 542 | 16.8% | | |
| | | | | One to two times a week | 207 | 542 | 38.2% | | |
| | | | | Almost every day or every day | 162 | 542 | 29.9% | | |
| Chersich (2014) *Cohort* | Kenya | HIV-negative FSWs | Snowball sampling | Alcohol abstinence | 144 | 399 | 36.1% | Lifetime abstinence or no alcohol use in the past 12 months | Moderate (7) |
| | | | | Low-risk drinking | 148 | 399 | 37.1% | AUDIT score 1-7 | |
| | | | | Hazardous drinking | 69 | 399 | 17.3% | AUDIT score 8-15 | |
| | | | | Harmful drinking | 38 | 399 | 9.5% | AUDIT score > 16 | |
| Coetzee (2018) *Cross sectional* | South Africa | FSWs | Respondent driven sampling | Problem drinking | 348 | 508 | 81.5% | AUDIT score of ≥3 | High (9) |
| | | | | Frequent and severe binge drinking | 278 | 508 | 54.7% | Adapted AUDIT score with a cut-off score of ≥6 (A new variable showing severe versus less severe/no binge drinking was created using the 3 original AUDIT-C items and the new volume variable.) | |
| Fawole (2014) *Cross sectional* | Nigeria | Brothel-based FSWs | Simple random sampling | Alcohol intake | 272 | 305 | 17.3% | Questionnaire | High (8) |
| Gezie (2015) *Cohort study* | Ethiopia | FSWs | Random sampling | Problem drinking | 115 | 474 | 24.26% | CAGE using cut off > 1 | High (9) |
| Goldenberg (2016) *Cross sectional* | Uganda | FSWs | Outreach | Worked under the influence of alcohol/drugs in the previous 6 months | 256 | 400 | 64.0% | Questionnaire | Moderate (6) |

(*Continued*)

**Table 1.** (Continued)

| Author, year & study design | Country | Sample | Sampling Procedure | Outcome(s) of interest | Events | Sample size | Event rate (%) | Method of assessing outcome (s) | Research quality |
|---|---|---|---|---|---|---|---|---|---|
| Kiene (2019) *Cross sectional* | Uganda | CSWs | Snowball sampling | Hazardous drinking and alcohol problems | 13 | 75 | 17.3% | AUDIT using cut-off > 7 | Moderate (6) |
| | | | | Hazardous drinking | 4 | 75 | 4% | AUDIT using cut-off > 3 for the first three items of the AUDIT | |
| | | | | Alcohol problems | 3 | 75 | 3% | AUDIT using cut-off > 3 for the final seven items of the AUDIT | |
| Lancaster (2016)[ii] *Cross sectional* | Malawi | HIV-infected FSWs | Venue-based sampling | Alcohol use prior to last vaginal sex with client | 41 | 138 | 30% | Behavioural survey | Moderate (6) |
| | | | | Hazardous drinking | 39 | 138 | 28.3% | AUDIT score 7-15 | |
| | | | | Harmful drinking | 17 | 138 | 28.3% | AUDIT score 16-19 | |
| | | | | Alcohol dependence | 20 | 138 | 14.5% | AUDIT score > 20 | |
| Lancaster (2017)[ii] *Cross sectional* | Malawi | HIV-infected FSWs | Venue-based sampling | Hazardous drinking | 29 | 96 | 30% | AUDIT score 7-15 | Moderate (6) |
| | | | | Harmful drinking | 10 | 96 | 10% | AUDIT score 16-19 | |
| | | | | Alcohol dependent | 11 | 96 | 12% | AUDIT score > 20 | |
| Leddy (2018) *Cross sectional* | Tanzania | FSWs | Venue-based time location sampling | Frequent intoxication during sex work in the past 30 days | 207 | 496 | 42.0% | Survey | Moderate (5) |
| | | | | Drink one or more drinks on a typical day of work | 408 | 496 | 97.0% | | |
| Nouaman (2015) *Cross sectional* | Côte d'Ivoire | FSWs | Convenience sampling | Moderate alcohol user | 84 | 249 | 33.7% | AUDIT score <8 | Moderate (7) |
| | | | | Hazardous alcohol user | 49 | 249 | 19.7% | AUDIT score ≥8 | |
| Ochonye (2019) *Cross sectional* | Nigeria | FSWs | Snowball sampling | Drank alcohol in the last 4 weeks | | | | Semi-structured interviewer-administered questionnaire | Moderate (7) |
| | | | | *Everyday* | 53 | 188 | 28.2% | | |
| | | | | *At least once a week* | 31 | 188 | 17.0% | | |
| | | | | *Occasionally* | 46 | 188 | 25.3% | | |
| Odukoya (2013) *Cross sectional* | Nigeria | FSWs | Venue-based sampling | Alcohol use | | | | Pretested structured questionnaire | Moderate (7) |
| | | | | *Current alcohol user* | 219 | 323 | 67.8% | | |
| | | | | *Ex-alcohol user* | 24 | 323 | 7.4% | | |
| | | | | Most recent drink | | | | | |
| | | | | *Less than a week ago* | 196 | 219 | 89.5% | | |
| | | | | *A week to a month ago* | 14 | 219 | 6.4% | | |
| | | | | *More than a month ago* | 9 | 219 | 4.1% | | |
| | | | | Amount of alcohol consumed per week in standard units | | | | | |
| | | | | *1-50* | 88 | 219 | 40.2% | | |
| | | | | *51-100* | 63 | 219 | 28.8% | | |
| | | | | *101-150* | 35 | 219 | 16% | | |
| | | | | *151-200* | 5 | 219 | 2.2% | | |
| | | | | *Above 200* | 28 | 219 | 12.8% | | |
| | | | | Age at first drink | | | | | |
| | | | | *<18 years* | 55 | 243 | 22.6% | | |
| | | | | *>18 years* | 188 | 243 | 77.4% | | |
| | | | | Level of drinking | | | | | |
| | | | | *Within 14 units of alcohol per week* | 24 | 219 | 11.0% | | |
| | | | | *Above 14 units of alcohol per week* | 195 | 219 | 89.0% | | |
| Parcesepe (2016)[iii] *Randomised control trial* | Kenya | FSWs-substance using | Non-probabilistic sampling | Hazardous drinking | 528 | 818 | 64.6% | AUDIT score 7-15 | Moderate (7) |
| | | | | Harmful drinking | 290 | 818 | 35.5% | AUDIT score 16-19 | |

(*Continued*)

**Table 1.** (Continued)

| Author, year & study design | Country | Sample | Sampling Procedure | Outcome(s) of interest | Events | Sample size | Event rate (%) | Method of assessing outcome(s) | Research quality |
|---|---|---|---|---|---|---|---|---|---|
| L´Engle (2014)[iii] *Randomised control trial* | Kenya | FSWs-substance using | Non-probabilistic sampling | Hazardous drinking | 529 | 818 | 65% | AUDIT score 7-15 | Moderate (7) |
| | | | | Harmful drinking | 290 | 818 | 35.5% | AUDIT score 16-19 | |
| Richter (2013) *Cross sectional* | South Africa | FSWs | Non-probabilistic sampling | Binge drinking; | | | | Binge drinking defined as having five or more alcohol drinks on one occasion. | Moderate (5) |
| | | | | *Daily* | 284 | 1566 | 18.1% | | |
| | | | | *Weekly* | 408 | 1566 | 26.1% | | |
| Wechsberg (2005)[iv] *Cross sectional* | South Africa | Substance-using FSWs | Non-probabilistic sampling | Alcohol use by age 17 | 47 | 93 | 51% | Self-reported; items not described in detail. | Moderate (6) |
| | | | | Alcohol use in the past 30 days; | | | | | |
| | | | | *Daily* | 17 | 93 | 18% | | |
| | | | | *At least twice a week* | 22 | 93 | 24% | | |
| Wechsberg (2006)[iv] *Cross sectional* | South Africa | FSWs who self-report cocaine use or have a positive urine test for cocaine | Non-probabilistic sampling | Alcohol use by age 17 | 47 | 93 | 51% | Self-reported; items not described in detail. | Moderate (6) |
| | | | | Alcohol use in past 30 days: | | | | | |
| | | | | *Daily* | 17 | 93 | 18% | | |
| | | | | *At least twice a week* | 22 | 93 | 24% | | |
| Wechsberg (2011) *Randomised clinical trial* | South Africa | FSWs who use substances | Non-probabilistic sampling | Days of drinking in past 30 days | 13.4 days (s. d. 9.8) | 550 | - | Self-reported use based on questionnaire | Moderate (7) |
| | | | | Days drunk in past 30 days | 10.5 days (s. d. 9.2 | 550 | - | Self-reported use based on questionnaire | |
| | | | | On drinking day, how many drinks on average, in past 30 days | 9.2 days (s. d. 11.0 | 550 | - | Self-reported use based on questionnaire | |
| Wechsberg (2008) *Cross sectional* | South Africa | FSWs who use substances | Targeted sampling | Alcohol prior to sex | 91 | 163 | 55.8% | Questionnaire. Item not described in detail. | Moderate (5) |
| Wechsberg (2009) *Cross sectional* | South Africa | FSWs who use substances | Targeted sampling | Lifetime alcohol use | 335 | 335 | 100% | Questionnaire based on DSM-IV | Moderate (6) |
| | | | | Lifetime alcohol use disorder: | | | | | |
| | | | | Abuse | 270 | 335 | 80.6% | | |
| | | | | Dependence | 232 | 335 | 69.3% | | |
| | | | | Abuse/dependence | 287 | 335 | 85.7% | | |
| | | | | Past year alcohol use disorder: | | | | | |
| | | | | Abuse | 252 | 335 | 75.2% | | |
| | | | | Dependence | 215 | 335 | 64.2% | | |
| | | | | Abuse or dependence | 274 | 335 | 81.8% | | |
| Weiss (2016) *Cohort* | Uganda | FSWs | Non-probabilistic sampling | Problem drinking | 572 | 1027 | 56.0% | CAGE using cut-off > 2 | Moderate (6) |
| Wilson (2016)[v] *Cross sectional* | Kenya | HIV-positive FSWs | Non-probabilistic sampling | Alcohol use problems | | | | | Moderate (6) |
| | | | | *Minimal* | 103 | 357 | 28.9% | AUDIT score 1-6 | |
| | | | | *Moderate* | 57 | 357 | 15.9% | AUDIT score 7-15 | |
| | | | | *Severe/possible AUD* | 14 | 357 | 3.9% | AUDIT score 16 or higher | |
| White (2016)[v] *Cohort* | Kenya | HIV-positive FSWs | Non-probabilistic sampling | *Low risk* | 116 | 405 | 28.6% | AUDIT score 1-6 | Moderate (7) |
| | | | | *Hazardous or harmful* | 88 | 405 | 21.7% | AUDIT score 7-40 | |
| Yadav (2005) *Cross sectional* | Kenya | HIV-negative FSWs | Random sampling | Daily alcohol use | 222 | 466 | 53.4% | Behavioral questionnaire | High (8) |

*(Continued)*

**Table 1.** (Continued)

| Author, year & study design | Country | Sample | Sampling Procedure | Outcome(s) of interest | | Events | Sample size | Event rate (%) | Method of assessing outcome (s) | Research quality |
|---|---|---|---|---|---|---|---|---|---|---|
| Fearon (2019) *Cross sectional* | Zimbabwe | HIV-negative FSWs | Respondent-driven sampling | Alcohol consumption over the past 12 month | | | | | Behavioral questionnaire | Moderate (7) |
| | | | | *Never* | | 262 | | 42.9% | | |
| | | | | *Once a month or less* | | 44 | 611 | 7.2% | | |
| | | | | *2-4 times/month* | | 77 | 611 | 12.6% | | |
| | | | | *2-3 times/week* | | 112 | | 18.3% | | |
| | | | | *4 or more times a week* | | 115 | 611 | 18.8% | | |
| | | | | Had more than 6 alcoholic drinks in one night during last 12 months | | | | | | |
| | | | | *Never – no alcohol last 12 months* | | 262 | 611 | 42.9% | | |
| | | | | *Never – drank alcohol but no occasions of more than 6 drinks* | | 169 | 611 | 27.7% | | |
| | | | | *Yes at least one occasion* | | 178 | 611 | 29.1% | | |
| **MIDDLE EAST AND NORTH AFRICA** | | | | | | | | | | |
| Kabbash (2012) *Cross sectional* | Egypt | FSWs | Random sampling | Alcohol intake in the last month: | | | | | Questionnaire | Moderate (5) |
| | | | | *Daily* | | 30 | 431 | 7.0% | | |
| | | | | *At least once weekly* | | 70 | 431 | 16.2% | | |
| | | | | *2-3 times monthly* | | 72 | 431 | 16.7% | | |
| Karamouzian (2017) *Cross sectional* | Iran | FSWs | Facility-based sampling | Alcohol use ever | | 466 | 451 | 54.5% | Survey | Moderate (5) |
| | | | | Alcohol use before sex ever | | 246 | 520 | 4.7% | | |
| **SOUTH ASIA** | | | | | | | | | | |
| Todd (2010) *Cross sectional* | Afghanistan | FSWs | Venue-based sampling | Alcohol use | | 26 | 520 | 4.7% | Questionnaire | Moderate (5) |
| | | | | Consuming 3 drinks or less each week | | 24 | 26 | 93.3% | | |
| | | | | Using alcohol or drugs with clients | | 27 | 51 | 53.9% | | |
| Barua (2012) *Cross sectional* | India | FSWs | Respondent driven sampling | Consumption of alcohol | | 304 | 426 | 71.4% | Questionnaire; items not described in more detail. | Moderate (5) |
| Bowen (2011) *Cross sectional* | India | FSWs | Convenience and snowball sampling | Alcohol use around time of first sex work | | 126 | 220 | 57.3% | Cross-sectional survey | Low (4) |
| Devine (2010) *Cross sectional* | India | FSWs | Outreach sampling | Regular AOD use at the time of first sex-work | | 108 | 186 | 58.1% | Questionnaire | Moderate (6) |
| | | | | Alcohol use around the time of first sex-work | | 101 | 186 | 54.3% | | |
| Heylen (2019) *Cross sectional* | India | FSWs | Non-probability sampling, including referrals from NGOs, brokers or other FSWs | Alcohol use frequency | *<1 time per week* | 99 | 589 | 16.8% | "On average, how often do you have a drink?" | Moderate (5) |
| | | | | | *1-2 days per week* | 238 | 589 | 40.4% | | |
| | | | | | *3-4 day per week* | 100 | 589 | 17.0% | | |
| | | | | | *5-6 days per week* | 9 | 589 | 1.5% | | |
| | | | | | *Every day* | 46 | 589 | 7.8% | | |
| Iaisuklang (2017) *Cross sectional* | India | FCSWs | Purposive Sampling | Intake of alcohol | | 79 | 100 | 79.0% | Sociodemographic data sheet | Low (3) |
| | | | | Alcohol dependence | | 8 | 100 | 8.0% | | |
| Pandiyan (2012) *Cross sectional* | India | FCSWs | Recruitment from hospital | Alcohol use | | 100 | 100 | 100.0% | Questionnaire | Low (2) |

*(Continued)*

**Table 1.** (Continued)

| Author, year & study design | Country | Sample | Sampling Procedure | Outcome(s) of interest | | Events | Sample size | Event rate (%) | Method of assessing outcome (s) | Research quality |
|---|---|---|---|---|---|---|---|---|---|---|
| Patel (2015) *Cross sectional* | India | FSWs | Conventional cluster sampling and time location cluster sampling | Alcohol use past 30 days | | 948 | 1986 | 47.7% | Questionnaire; item not described in further detail | Moderate (7) |
| Sagtani (2013) *Cross sectional* | Nepal | FSWs | Snowball sampling | Previous history of sexual intercourse under influence of alcohol | | 96 | 200 | 45.7% | Semi-structured questionnaire. Items not described in detail. | Moderate (6) |
| Samet (2010) *Cross sectional* | India | HIV-infected FSWs | Purposive sampling | Any alcohol use in the last 30 days | | 80 | 211 | 38% | The number who said they had had any alcohol in the last 30 days. | Moderate (5) |
| | | | | Heavy alcohol use | | 67 | 211 | 32% | Using cut-off >3 drinks in a day or >7 drinks/week. | |
| | | | | Alcohol dependence | | 23 | 211 | 11% | CIDI | |
| Singh (2016) *Cross sectional* | India | FSWs | Convenience sampling | Drinking | | 33 | 120 | 27.5% | Item not described clearly | Moderate (6) |
| Verma (2010) *Cross sectional* | India | Migrant FSWs | Two-stage random sampling | Alcohol use in the last 1 month | *Any alcohol intake* | 2115 | 3412 | 62.0% | Survey | High (8) |
| | | | | | *All types of alcohol* | 662 | 3412 | 19.4% | | |
| | | | | | *Alcohol use prior to sex* | 1853 | 3412 | 53.8% | | |
| EUROPE AND CENTRAL ASIA | | | | | | | | | | |
| Wirtz (2015) *Cross sectional* | Russia | FSWs | Respondent driven sampling | Alcohol use while selling sex last 6 months | | 523 | 754 | 69.4% | Questionnaire; item not described in further detail | Moderate (5) |
| Davis (2017) *Cross sectional* | Kazakhstan | HIV-positive FSWs | Recruitment through referrals | Hazardous drinking | | 23 | 56 | 41.1% | AUDIT score >3 | Moderate (7) |
| EAST ASIA and PACIFIC | | | | | | | | | | |
| Nemoto (2013) *Cross sectional* | Thailand | FSWs | Purposive sampling | Alcohol use in the past 12 months | | 192 | 205 | 93.7% | Questionnaire with open-ended questions; items not described more clearly. | Moderate (5) |
| | | | | Having sex with customers under the influence of alcohol in the past 6 months | | 165 | 205 | 80.4% | | |
| | | | | Having sex with primary partners under the influence of alcohol in the past 6 months | | 78 | 121 | 64.5% | | |
| Chen (2013)[vi] *Cross sectional* | China | FSWs | Convenience sampling | Probable drinking problem | | 343 | 686 | 50% | AUDIT using cut-off >8 | Moderate (7) |
| | | | | Probably alcohol dependence | | 182 | 686 | 27% | AUDIT using cut-off >13 | |
| | | | | Risk drinking | | 217 | 686 | 32% | AUDIT score 8-15 | |
| | | | | Heavy drinking | | 78 | 686 | 11.3% | AUDIT score 16-19 | |
| | | | | Hazardous drinking | | 48 | 686 | 7% | AUDIT score 20-40 | |
| Chen (2015)[vi] *Cross sectional* | China | FSWs | Convenience sampling | Risk drinking | | 322 | 1022 | 31.5% | AUDIT score 8-15 | Moderate (7) |
| | | | | Heavy drinking | | 119 | 1022 | 11.6% | AUDIT score 16-19 | |
| | | | | Hazardous drinking | | 88 | 1022 | 8.6% | AUDIT score 20-40 | |

(*Continued*)

**Table 1.** (Continued)

| Author, year & study design | Country | Sample | Sampling Procedure | Outcome(s) of interest | | Events | Sample size | Event rate (%) | Method of assessing outcome (s) | Research quality |
|---|---|---|---|---|---|---|---|---|---|---|
| Couture (2016) *Cross sectional* | Cambodia | FSWs | Convenience sampling | AUDIT-C (last 3 months): | *Abstinence or lower risk use* | 15 | 100 | 15.0% | AUDIT-C score 0-2 | Moderate (6) |
| | | | | | Unhealthy alcohol use | 85 | 100 | 85.0% | AUDIT-C score 3-12 | |
| | | | | Alcohol use frequency (last 3 months): | Never/less than once a month | 60 | 100 | 60.0% | questionnaire | |
| | | | | | 2-4 times a month | 10 | 100 | 10.0% | | |
| | | | | | 2 or more times a week | 30 | 100 | 30.0% | . | |
| | | | | Number of drinks on a typical day: | 1-2 drinks | 7 | 96 | 7.3% | | |
| | | | | | 3-4 drinks | 13 | 96 | 13.5% | | |
| | | | | | 5 or more drinks | 76 | 96 | 79.2% | | |
| | | | | Being drunk or intoxicated recently (last 3 months) | | 81 | 100 | 81.0% | | |
| | | | | Heavy episodic drinking (last 3 months) | | 83 | 100 | 83.0% | | |
| | | | | Regular heavy drinking | | 29 | 100 | 29.0% | | |
| Fang (2007)[vii] *Cross sectional* | China | FSWs | Ethnographic targeted sampling | Sex w/alcohol | | 133 | 454 | 29.2% | Self-administered questionnaire | Low (4) |
| | | | | Alcohol intoxication last 6 months | | 149 | 454 | 32.8% | | |
| Hong (2007)[vi] *Cross sectional* [i] | China | FSWs | Outreach | Alcohol intoxication in past 6 months | | 149 | 454 | 32.8% | "Have you gotten drunk at least once a month in the past 6 months?" | Moderate (5) |
| Le (2019) *Cross sectional* | Vietnam | FSWs | Time-location sampling | Daily alcohol consumption in past month | | 223 | 1861 | 12.0% | Cross-sectional survey | Moderate (6) |
| Nemoto (2008) *Cross sectional* | Vietnam | FSWs | Stratified sampling | Alcohol use in the past 12 months | | 121 | 136 | 89.0% | Survey interview; items not described in detail. | Moderate (6) |
| Liao (2012) *Cross sectional* | China | FSWs | Respondent-driven sampling | Ever drinking alcohol | | 612 | 794 | 77.1% | Questionnaire | Low (4) |
| Parcesepe (2015) *Cross sectional* | Mongolia | Alcohol-using FSWs | Convenience sampling | Harmful alcohol use | | 7 | 222 | 3.2% | AUDIT score 8-15 | Moderate (7) |
| | | | | Hazardous alcohol use | | 9 | 222 | 4.1% | AUDIT score 16-19 | |
| | | | | Alcohol dependence | | 206 | 222 | 92.8% | AUDIT score >20 | |
| Su (2014) *Cross sectional* | China | FSWs | Venue based sampling | Mean AUDIT score | | 9.45 ±6.77 | 1022 | - | AUDIT score | Moderate (6) |
| Tran (2014) *Cross sectional* | Vietnam | FSWs | Peer-educator and staff referrals | Alcohol use | | 1663 | 1999 | 83.4% | Questionnaire | Low (4) |
| Urada (2012) *Cross sectional* | Philippines | FSWs | Venue based sampling | Used alcohol | | 99 | 142 | 70% | Questionnaire; item not described in more detail | Moderate (5) |

*(Continued)*

**Table 1.** (Continued)

| Author, year & study design | Country | Sample | Sampling Procedure | Outcome(s) of interest | | Events | Sample size | Event rate (%) | Method of assessing outcome(s) | Research quality |
|---|---|---|---|---|---|---|---|---|---|---|
| Urada (2014) *Cross sectional* | Philippines | FSWs | Venue-based sampling | Current alcohol use; Daily | | 71 | 482 | 14.7% | "How often do you have beer or drinks containing alcohol?" | Moderate (6) |
| | | | | Drinks alcohol, not daily | | 231 | 482 | 48% | | |
| | | | | Alcohol use with venue patron ever | | 165 | 482 | 34% | "How often do you drink beer or alcohol with your venue patron?" | |
| | | | | Alcohol intoxicated during sex ever | | 159 | 482 | 32% | "How often are you drunk when you have sex?" | |
| Witte (2010) *Cross sectional* | Mongolia | FSWs | Purposive sampling | Alcohol consumption on a typical day | | 44 | 48 | 92% | Items not described in detail. | Low (3) |
| | | | | 1-2 drinks/day | | 9 | 48 | 29% | | |
| | | | | 3-4 drinks/day | | 14 | 48 | 29% | | |
| | | | | 5-6 drinks/day | | 13 | 48 | 27% | | |
| | | | | 7-9 drinks/day | | 2 | 48 | 4% | | |
| | | | | 10 or more drinks/day | | 6 | 48 | 13% | | |
| | | | | Being unable to stop drinking once they have started on at least a monthly basis | | 39 | 48 | 81% | | |
| Witte (2011) *Randomized clinical trial* | Mongolia | FSWs (alcohol using FSWs) | Non-probabilistic sampling | AUDIT score >8 | | 229 | 229 | 100% | AUDIT | Moderate (6) |
| Zhang (2017) *Cross sectional* | China | FSWs | Non-probability sampling | Average AUDIT score | | 8.4 | 673 | - | AUDIT | Low (4) |
| Zhang (2014)[viii] *Cross sectional* | China | FSWs | Venue based sampling | Alcohol intoxication | | 627 | 968 | 64.8% | Questionnaire: frequencies of alcohol intoxication (e.g. every day, once 2–3 days, once a week, once 2–3 weeks, never) | Moderate (5) |
| Zhang (2014)[viii] *Cross sectional* | China | FSWs | Venue based sampling | Mean audit score (sd) | | 9.05 (7.36) | 1022 | | AUDIT | Moderate (6) |

LATIN AMERICA AND THE CARIBBEAN

| Author, year & study design | Country | Sample | Sampling Procedure | Outcome(s) of interest | | Events | Sample size | Event rate (%) | Method of assessing outcome(s) | Research quality |
|---|---|---|---|---|---|---|---|---|---|---|
| Aguayo (2008) *Cross sectional* | Paraguay | FCSWs | Non-probabilistic sampling | Sexual intercourse under the effects of alcohol | Sometimes | 341 | 723 | 47.2% | Questionnaire; items not described. | Low (4) |
| | | | | | Always | 106 | 723 | 14.7% | | |
| Bautista (2006) *Cross sectional* | Argentina | FSWs | Non-probabilistic sampling | Use of alcohol | | 36 | 1782 | 2.0% | Questionnaire; item not described | Low (4) |
| Bazzi (2015) *Cohort study* | Mexico | FSWs | Targeted and snowball sampling | Hazardous/harmful alcohol drinking in the past 6 months | | 42 | 212 | 20.0% | AUDIT using cut-off <8 | Moderate (7) |
| Caetano (2013) *Cross sectional* | Brazil | FSWs | Respondent-driven sampling | Alcohol use last month | *Every day* | 144 | 395 | 36.4% | Standardized questionnaire adapted from the FSW module of the Health International Behavioral Surveillance Surveys | Moderate (5) |
| | | | | | *Three times a week* | 131 | 395 | 33.2% | | |
| | | | | | *At least once a month* | 49 | 395 | 12.4% | | |

*(Continued)*

**Table 1.** (Continued)

| Author, year & study design | Country | Sample | Sampling Procedure | Outcome(s) of interest | | Events | Sample size | Event rate (%) | Method of assessing outcome(s) | Research quality |
|---|---|---|---|---|---|---|---|---|---|---|
| Carrasco (2019) *Cross sectional* | Dominican Republic | HIV-positive FSWs | Non-probabilistic hybrid sampling | Alcohol use | | 83 | 223 | 37.2% | Alcohol use: Study participants were asked about alcohol use in the last 30 days. | Moderate (6) |
| Chen (2012) *Cross sectional* | Mexico | FSWs | Modified venue-based sampling | Heavy alcohol use | *1-4 drinks on typical drinking day* | 30 | 174 | 17.2% | Quantitative surveys | Moderate (6) |
| | | | | | *>4 drinks on typical drinking day* | 144 | 174 | 82.8% | | |
| | | | | Frequency of alcohol use | *Never-1x/ week* | 58 | 189 | 30.7% | | |
| | | | | | *Several times/ week- daily* | 131 | 189 | 69.3% | | |
| Conners (2016) *Cross sectional* | Mexico | FSWs | Modified time-location sampling | Use alcohol often or always before sex | | 117 | 496 | 24.0% | Survey | Moderate (7) |
| Costa Passos (2004) *Cross sectional* | Brazil | FSWs | Venue-based sampling and snowball sampling | Being under the influence of alcohol | Daily/weekly | 287 | 462 | 62.1% | Questionnaire | Moderate (6) |
| | | | | | Rare/never | 175 | 462 | 37.9% | | |
| Dal Pogetto (2012) *Cross sectional* | Brazil | FSWs | Non-probabilistic sampling | Use alcohol at work | | 86 | 102 | 84.3% | Interview | Moderate (6) |
| Damacena (2014) *Cross sectional* | Brazil | FSWs | Respondent-driven sampling | Frequency of alcohol consumption | *Moderate (around once a week or less)* | 1160 | 2523 | 46.0% | Questionnaire in an Audio Computer-Assisted Self-Interview | Moderate (6) |
| | | | | | *Elevated (several times a week or every day)* | 700 | 2523 | 27.7% | | |
| de Matos (2017) *Cross sectional* | Brazil | FSWs | Respondent-driven sampling | Alcohol use in the last month | *No/only once a month* | 95 | 293 | 21.5% | Questionnaire; items not described. | Moderate (7) |
| | | | | | *At least three times a week* | 103 | 293 | 21.0% | | |
| | | | | | *Everyday* | 95 | 293 | 22.8% | | |
| Devoglio (2017) *Cross sectional* | Brazil | FSWs | Non-probabilistic sampling | Consumes alcohol | 74 | 83 | 89.2% | Questionnaire | Moderate (7) | |
| Donastorg (2014) *Cross sectional* | Dominican Republic | HIV-positive FSWs | Non-probabilistic hybrid sampling | Alcohol use in last 30 days | *At least once a week* | 125 | 298 | 35.4% | Socio-behavioral survey | Moderate (5) |
| | | | | | *Less than weekly* | 173 | 298 | 64.6% | | |
| | | | | Alcohol use before sex *Sometimes/Always* | | 125 | 267 | 46.8% | | |
| Duncan (2010) *Cross sectional* | Jamaica | FSWs | Random sampling | Everyday alcohol use | | 264 | 450 | 58.7% | Questionnaire | Low (3) |
| Gaines (2013) *Cohort* | Mexico | FSW-IDUs | Outreach | Weekly alcohol consumption | | 248 | 567 | 43.7% | Face-to-face interviews | Low (4) |

*(Continued)*

**Table 1.** (Continued)

| Author, year & study design | Country | Sample | Sampling Procedure | Outcome(s) of interest | | Events | Sample size | Event rate (%) | Method of assessing outcome (s) | Research quality |
|---|---|---|---|---|---|---|---|---|---|---|
| Goldenberg (2012) *Cross sectional* | Mexico | FSWs | Outreach | Used alcohol before starting sex work | | 403 | 624 | 64.4% | Questionnaire | Moderate (5) |
| Hooi (2018) *Cross sectional* | Curacao | FSWs | Random sampling | Alcohol use | | 29 | 76 | 39.2% | Survey | Low (4) |
| Jain (2018) *Cross sectional* | Mexico | FSW-IDUs | Random sampling | Alcohol use before or during sex with clients | | 296 | 584 | 50.8% | Survey | Moderate (5) |
| | | | | Binge drinking - Five or more alcoholic beverages in one sitting in the past months | | 271 | 584 | 46.5% | | |
| Jain (2020) *Cross sectional* | Mexico | FSWs | Random sampling | Hazardous alcohol consumption in the past year | | 136 | 295 | 46.1% | AUDIT using cut-off >8 | Moderate (6) |
| Kerrigan (2016) *Cross sectional* | Dominican Republic | FSWs | Hybrid sampling; recruitment by FSW peer navigators and referrals | Alcohol use in the last 30 days | *At least once a week* | 123 | 228 | 53.9% | Interviewer-administered socio-behavioral survey | Moderate (7) |
| | | | | | *Less than once a week* | 105 | 228 | 46.1% | | |
| | | | | Alcohol use before sex *Sometimes/always* | | 108 | 228 | 47.4% | | |
| Munoz (2006) *Cross sectional* | Venezuela | FCSWs | Time-location sampling | Alcohol consumption; | 1 or 2 drinks/ month | 104 | 613 | 17.0% | Questionnaire; items not described. | Lowe (4) |
| | | | | | 1 drink/week | 188 | 613 | 30.7% | | |
| | | | | | 1 drink/day | 56 | 613 | 9.1% | | |
| | | | | | >1 drink/day | 73 | 613 | 11.9% | | |
| Munoz (2010) *Cross sectional* | Mexico | FSWs | Recruitment through outreach | Used alcohol in the last month | | 673 | 924 | 73.0% | Baseline survey face-to-face | Moderate (6) |
| | | | | Used alcohol during or before sex work | | 546 | 924 | 59.0% | | |
| Pando (2006) *Cross sectional* | Argentina | FCSWs | Snowball sampling | Alcohol consumption; | <Once a week | 486 | 625 | 77.8% | Standardised questionnaire; items not described in detail. | Moderate (6) |
| | | | | | >Once a week | 139 | 625 | 22.2% | | |
| Persaud (2000) [i] *Cross sectional* | Guyana | FCSWs | Non-probabilistic sampling | Always under the influence of alcohol while having sex with their last 10 clients | | 43 | 124 | 34.5% | Questionnaire | Low (3) |
| Persaud (1999) [vii] *Cross sectional* | Guyana | FCSWs | Non-probabilistic sampling | Regular alcohol consumption while looking for clients | | 119 | 124 | 88.0% | Questionnaire | Low (3) |
| Salazar (2019) *Cross sectional* | Mexico | FSWs | Time-location sampling | First month alcohol use | | 402 | 603 | 66.0% | Questionnaire | Moderate (5) |
| | | | | First month forced alcohol use | | 26 | 650 | 4.0% | | |

*(Continued)*

**Table 1.** (Continued)

| Author, year & study design | Country | Sample | Sampling Procedure | Outcome(s) of interest | Events | Sample size | Event rate (%) | Method of assessing outcome(s) | Research quality |
|---|---|---|---|---|---|---|---|---|---|
| Semple (2016)[ii] *Cross sectional* | Mexico | HIV-negative FSWs | Time-location sampling | Hazardous drinking | 835 | 1089 | 76.7% | AUDIT-C using cut-off >3 | Moderate (6) |
| | | | | Used alcohol before or during sex with client | 651 | 1089 | 65.0% | "Have you used alcohol before or during sex with client in the past month?" | |
| Semple (2015)[viii] *Cross sectional* | Mexico | HIV-negative FSWs | Time-location sampling | Used alcohol before or during sex with client(s) in past month | 661 | 1089 | 60.7% | Participants were asked how often in the past month they had used alcohol before or during sex with a client. Response categories (never, sometimes, often, always) were recoded yes/no to create two dichotomous variables | Moderate (6) |
| Semple (2017)[viii] *Cross sectional* | Mexico | HIV-negative FSWs | Time-location sampling | Hazardous alcohol use | 835 | 1089 | 76.7% | AUDIT-C using cut-off >3 | Moderate (6) |
| | | | | Used alcohol with client in past month | 661 | 1089 | 60.7% | Item not described in detail | |
| Servin (2017) *Cross sectional* | Mexico | FSWs | Time-location sampling | Always used alcohol right before or during sex with clients in the past 30 days | 381 | 603 | 63.7% | Questionnaire | Moderate (5) |
| Strathdee (2008)[iii] *Cross sectional* | Mexico | FSWs | Venue based sampling | Often/always used alcohol before/during vaginal sex | 207 | 924 | 22.4% | Questionnaire; items not described in detail | Moderate (6) |
| Patterson (2006)[ix] *Cross sectional* | Mexico | FSWs | Recruitment through health clinics, street outreach and referrals. | Alcohol use in the past month | 273 | 295 | 93.0% | Interviewer-administered survey | Moderate (5) |
| Ulibarri (2014)[ix] *Cross sectional* | Mexico | FSWs | Non-probabilistic sampling | Used alcohol in the past month | 673 | 924 | 72.8% | Interview; items not described in detail. | Moderate (5) |
| | | | | Used alcohol before sex with clients | 297 | 924 | 32.1% | | |

[i] Bitty-Anderson (2019) and Tchankoni (2020)–papers report on same study

[ii] Lancaster (2016) and Lancaster (2017)–papers report on same study

[iii] L'Engle (2014) and Parcesepe (2016)–papers report on same study

[iv] Wechsberg (2006) and Wechsberg 2005 –papers report on same study

[v] Wilson (2016) and White (2016)–papers report on same study

[vi] Chen (2013) and Chen (2015)–papers report on same study

[vii] Hong (2007) and Fang (2007)–papers report on same study

[viii] Zhang (2014) and Zhang (2014)–papers report on same study

[ix] Persaud (2000) and Persaud (1999)–papers report on same study

associations with the outcome. Studies that met inclusion criteria were organised by similar findings and reported through a narrative synthesis. Prevalence estimates were calculated from percentages or proportions. Meta-analyses were conducted to estimate global or regional prevalence on studies that scored moderate or high in quality assessment. Through narrative synthesis of available measures (e.g. frequency of alcohol use) and known validated tools (e.g. AUDIT, CAGE) we selected a range of measures of alcohol use and harmful alcohol use for meta-analyses. Where results from a single study were reported in multiple papers, we

**Table 2. Associations with alcohol use (cross-sectional studies).**

| Author & Study | Country | Sample | Alcohol use measure | Outcome of interest | Sample size | Odds in the exposed[1] | Odds in the unexposed[2] | Crude Odds Ratio (95% CI) | P-value |
|---|---|---|---|---|---|---|---|---|---|
| **Violence and arrest** | | | | | | | | | |
| Chersich (2014)* | Kenya | HIV-negative FSWs | Hazardous/ Harmful drinking (AUDIT score ≥ 8) | Sexual violence (physically forced to have sex) | 399 | 39/68 | 32/260 | 4.66 (2.72–7.98) | <0.001 |
| | | | | Physical violence | 399 | 77/30 | 59/233 | 10.13 (6.09–16.87) | <0.001 |
| Jain (2020) | Mexico | FSWs | AUDIT using cut-off >8 | Lifetime experience of physical abuse or sexual violence perpetrated by a client. | 295 | 58/49 | 78/110 | 1.67 (1.04–2.69) | 0.04 |
| Semple (2016) | Mexico | FSWs | AUDIT-C using cut-off >3 | Forced to have sex with client in past year | 1001 | 118/717 | 27/139 | 0.85 (0.54–1.34) | 0.5 |
| Semple (2016) | Mexico | FSWs | AUDIT-C using cut-off >3 | Ever arrested | 1001 | 176/659 | 39/127 | 0.87 (0.59, 1.29) | 0.5 |
| **Condom Use** | | | | | | | | | |
| Weiss (2016) | Uganda | FSWs | CAGE using cut-off > 2 | Inconsistent condom use (time period not specified) | 905 | 218/298 | 146/243 | 1.22 (0.93–1.59) | 0.2 |
| Chen (2013) | China | FSWs | AUDIT using cut-off >8 | Inconsistent condom use with stable partner most recent 3 episodes of sex | 983 | 285/260 | 182/256 | 1.54 (1.20–1.99) | 0.01 |
| | | | | Inconsistent condom use with casual partner most recent 3 episodes of sex | 983 | 207/338 | 90/348 | 2.37 (1.77–3.16) | <0.001 |
| Semple (2016) | Mexico | HIV negative FSWs | AUDIT-C using cut-off >3 | Unprotected vaginal and anal sex acts with clients | 1001 | 131/704 | 28/138 | 1.04 (0.73–1.49) | 0.7 |
| **Other sexual risk behaviours** | | | | | | | | | |
| Bukenya (2019) | Uganda | FSWs | Hazardous alcohol use (AUDIT Score ≥ 7) | Unplanned pregnancy | 819 | 319/143 | 214/143 | 1.5 (1.12–1.99) | 0.007 |
| Semple (2016) | Mexico | FSWs | AUDIT-C using cut-off >3 | Total number of clients | 1001 | 430/404 | 108/58 | 0.52 (0.37–0.74) | 0.002 |
| Chersich (2014) | Kenya | HIV-negative FSWs | Hazardous or harmful drinking (AUDIT score ≥ 8) | using any contraception (condoms/oral contraception, injectable or implant) | 399 | 64/43 | 189/103 | 0.81 (0.51–1.28) | 0.4 |
| **HIV/STI prevalence** | | | | | | | | | |
| Couture (2016) | Cambodia | FSWs | AUDIT-C score 3–12 | HIV prevalence | 100 | 4/81 | 5/10 | 0.1 (0.03–0.43) | 0.002 |
| Nouaman (2015) | Côte d'Ivoire | FSWs | AUDIT score ≥8 (hazardous) | HIV prevalence | 249 | 11/73 | 38/127 | 0.50 (0.24–1.05 | 0.07 |
| Weiss (2016) | Uganda | FSWs | CAGE using cut-off > 2 | HIV prevalence | 1027 | 225/347 | 156/299 | 1.24 (0.96–1.61) | 0.1 |
| | | | | HSV-2 | 1027 | 474/98 | 348/107 | 1.49 (1.09–2.02) | 0.01 |
| | | | | Active syphilis | 1027 | 61/508 | 42/413 | 1.18 (0.78–1.79) | 0.4 |
| | | | | Bacterial vaginosis | 1027 | 327/245 | 246/209 | 1.14 (0.89–1.45) | 0.3 |
| | | | | Candida ssp. | 1027 | 71/501 | 41/414 | 1.43 (0.95–2.15) | 0.08 |
| | | | | Trichomonas vaginalis | 1027 | 95/377 | 81/374 | 1.16 (0.84–1.62) | 0.4 |
| | | | | Neisseria gonorrhoeae | 1026 | 85/457 | 49/405 | 1.54 (1.06–2.24) | 0.03 |
| | | | | Chlamydia trachomatis | 1026 | 51/521 | 41/413 | 0.99 (0.64–1.52) | 0.9 |
| | | | | Mycoplasma genitalium | 1025 | 94/478 | 54/399 | 1.45 (1.01–2.08) | 0.04 |
| Chersich (2014) | Kenya | HIV-negative FSWs | Hazardous/harmful drinking (AUDIT score ≥ 8) | STI prevalence (syphilis or trichomonas) | 399 | 10/97 | 14/278 | 2.05 (0.88–4.76) | 0.1 |

*(Continued)*

**Table 2.** (Continued)

| Author & Study | Country | Sample | Alcohol use measure | Outcome of interest | Sample size | Odds in the exposed[1] | Odds in the unexposed[2] | Crude Odds Ratio (95% CI) | P-value |
|---|---|---|---|---|---|---|---|---|---|
| Chen (2013) | China | FSWs | AUDIT using cut-off >8 | History of STI | 983 | 46/499 | 28/410 | 1.35 (0.83–2.20) | 0.2 |
| Jain (2020) | Mexico | FSWs | AUDIT using cut-off >8 | Tested positive for chlamydia, gonorrhea, or active syphilis. | 295 | 27/107 | 27/132 | 1.33 (0.74, 2.37) | 0.3 |
| **Mental Health Problems and drug use** | | | | | | | | | |
| Jain (2020) | Mexico | FSWs | AUDIT using cut-off >8 | Polydrug use last month (Use of ≥2 illicit drugs (heroin; methamphetamine; cocaine; inhalants; ecstasy; tranquilizers; barbiturates) in the past month) | 295 | 44/92 | 41/118 | 1.38 (0.83–2.28) | |
| | | | | Moderate or severe depression defined as a score ≥20 on the Beck Depression Inventory II (BDI-II). | 295 | 57/79 | 49/110 | 1.62 (1.00–2.62) | 0.05 |
| Chersich (2014) | Kenya | HIV-negative FSWs | Hazardous or harmful drinking (AUDIT score ≥ 8) | Cannabis in past week | 399 | 17/90 | 13/279 | 4.05 (1.90–8.67) | 0.0003 |
| Chersich (2014) | Kenya | HIV-negative FSWs | Hazardous or harmful drinking (AUDIT score ≥ 8) | Khat use in past week | 399 | 35/72 | 39/253 | 3.15 (1.86–5.34) | <0.0001 |
| Semple (2016) | Mexico | FSWs | AUDIT-C using cut-off >3 | Used drugs in past month | 1001 | 99/736 | 7/159 | 3.06 (1.39–6.70) | <0.01 |
| Coetzee (2018) | South Africa | FSWs | Adapted AUDIT-C score with cut off of ≥6 (frequent and severe binge drinking) | Depression (20-item CES-D scale) | 508 | 196/81 | 174/55 | 0.76 (0.51–1.14) | 0.2 |
| | | | | PTSD (PTSD-8) | 508 | 99/179 | 96/134 | 0.77 (0.54–1.11) | 0.2 |
| Tchankoni (2020) | Togo | FSWs | AUDIT score > 7 | Psychological distress (Kessler) | 952 | 82/99 | 350/421 | 1.0 (0.7–1.4) | 1.0 |

*Cohort study but some associations reported at baseline.

[1] odds in the exposed (e.g. alcohol and violence/alcohol and no violence).

[2] odds in the unexposed (e.g. no alcohol and violence/no alcohol and no violence).

included all studies in Table 1 but only the original study was reported in the prevalence analyses. The global pooled estimates for the mean prevalence of alcohol use and its associated 95% confidence intervals (95% CI) were calculated in Stata using random effects models. Heterogeneity was measured through the Higgins' $I^2$ statistic, which gives the percentage of total variation across studies in a meta-analysis that is due to heterogeneity rather than chance, ranging from 0–100% [38]. A high $I^2$ statistic would indicate greater heterogeneity between studies. Analyses were conducted in STATA 16.1 (Stata Inc., College Station, TX, USA).

For all included studies, any associations with alcohol use were reported in narrative synthesis. Only studies using validated alcohol measurement tools were eligible to have associations data included in meta-analyses. Sub-group analyses examined associations between alcohol use (measured using a validated tool) and common health and social concerns for FSWs including HIV status, STIs, condom use, other drug use, violence experience and mental health.

# Results

## Study characteristics

The initial electronic search yielded 416 results, with 19 additional studies identified through citation searching. After duplicate records were removed, the titles and abstracts of 290

publications (271 from databases, 19 from other sources) were screened for eligibility. Of these, 140 full texts were identified as potentially relevant publications and reviewed for inclusion. Ninety-nine papers reporting on 87 unique studies with 51,904 participants met the inclusion criteria (Fig 1) with year of publication ranging from 1999 to 2020. Studies were based in 32 LMICs: 10 in Sub-Saharan Africa, 2 in the Middle East and North Africa, 3 in South Asia, 2 in Europe and Central Asia, 6 in East Asia and the Pacific and 9 in Latin America and the Caribbean.

**Study population.** The majority of sampling techniques were non-probabilistic, which included convenience (n = 10), purposive (n = 5), snowball (n = 9), venue-based (n = 10), respondent driven sampling (n = 8), outreach (n = 7), targeted (n = 3) and time-location (n = 9). Twenty eight studies reported non-probabilistic sampling with no further detail on sampling methods. Ten studies used probabilistic (simple random sampling) methods (Table 1). Ten studies selected participants based on harmful alcohol or drug use [39–48] and were included in Table 1 but excluded from the meta-analyses (regardless of CEBMa score) to avoid biasing the pooled estimates. Thirteen studies selected participants based on HIV status (n = 6 HIV negative [27, 28, 49–52], n = 8 HIV positive [53–60]) and results were reported with these studies included and excluded in meta-analysis for comparison.

**Study design and quality.** Eighty-nine studies used cross-sectional designs to report on alcohol use, six used a cohort design [27, 33, 47, 55, 61, 62] and four experimental studies reported an alcohol intervention [39, 40, 43, 48]. Five studies scored as high quality, 79 scored as moderate and 15 scored as weak quality (Table 1 and S3 Appendix).

## Patterns of alcohol use

The results on patterns of alcohol use are shown in Table 1. The majority of FSWs drank frequently, though drinking patterns varied with setting and tools used to measure alcohol use. A variety of measures on drinking prevalence and frequency were reported with the majority of studies (75.6%, n = 65 studies) not using a validated tool to assess alcohol use. Overall, 2.0–100.0% of FSWs reported consuming any alcohol (no timeframe) [59, 63–80].

For studies reporting on alcohol use frequency, 12.0–100% used alcohol in the past month [41–43, 50–52, 57, 60, 63, 80–92]; 89.0–93.7% reported using alcohol in the last 12 months [93, 94]; 6.4–77.8.0% used alcohol at least once a week [29, 41, 42, 52, 60, 63, 85, 87–90, 95–102] and 10.0–64.6% used alcohol at least once a month [29, 52, 60, 63, 85, 87, 89, 90, 97].

In terms of drinking patterns, 32.8–81.0% reported being drunk when using alcohol [43, 71, 82, 84, 97, 103–105] and 26.1–54.7% reported binge drinking [52, 63, 83, 95, 106–108].

Alcohol use was common during sex with 32.1–97.0% reporting having used alcohol during sex work [28, 50, 51, 76, 84, 89, 91, 92, 106, 108–115]; 22.4–80.4% reported having any sexual intercourse (with clients/partners or unspecified) under the influence of alcohol [28, 44, 50, 53, 60, 71, 79, 80, 89, 92, 93, 103, 108, 114–119] and 54.3–66.9% used alcohol at the time of first sex work [111, 112, 120, 121].

## Prevalence of alcohol use reported using validated tools

In total, 29 papers from 22 unique studies reported on prevalence of alcohol use using validated tools [27, 28, 32, 33, 39, 40, 46, 48, 51, 53–55, 57, 58, 61, 63, 83, 84, 97, 107, 113, 122–129] (Table 1). Studies were based in thirteen LMICs including 7 in sub-Saharan Africa, 1 in South Asia, 1 in Europe and Central Asia, 4 in East Asia and Pacific and 1 in Latin America. The majority of these studies were cross sectional (n = 25), while two studies were cohort [27, 33] and two were randomised controlled trial [40, 48]. Tools and cut off scores to measure alcohol use disorders varied. In total 26 studies [27, 28, 32, 39, 40, 46, 48, 51, 53–55, 58, 62, 83, 84, 97,

107, 113, 122–129] used the AUDIT tool—three studies reported on mean AUDIT score [127–129], five [28, 51, 58, 97, 123] used the shortened AUDIT-C with score ≥3 as a cut-off and one an adapted AUDIT-C [107]. Of the sixteen studies using the full AUDIT tool, nine used AUDIT cut off ≥7 [29, 39, 40, 53–56, 126, 127] to define hazardous alcohol use and eight used AUDIT cut off ≥8 [27, 32, 46, 48, 62, 113, 122, 126]. Two studies used CAGE [33, 61] with one study using a cut-off of ≥2 and one study using a cut-off of ≥1, and one study used the WHO CIDI tool [57].

## Harmful alcohol use

A meta-analysis was conducted with seventeen unique studies that used a validated tool (excluding studies that included only substance using FSWs) and estimated the pooled prevalence of any harmful alcohol use to be 41% (31–51%) ($I^2$ = 98.87%) (Fig 2). The same analysis was conducted without the six studies that had selected participants based on HIV-status (as this could potentially bias the findings) and this analysis estimated the pooled prevalence to be similar (43% (95% CI: 31–55%) ($I^2$ = 98.96%)). The pooled prevalence of harmful or dependent alcohol use only (AUDIT ≥16/CIDI tool) was 14% (95% CI: 6–22%)($I^2$ = 97.05%) (Fig 3). Prevalence estimates were conducted for each of the different regions were there was appropriate data, resulting in an estimated pooled prevalence of harmful alcohol use to be 38% (95% CI: 27–48%)($I^2$ = 98.40%) in Sub-Saharan Africa (Fig 4); 47% (95% CI: 17–77%) ($I^2$ = 99.62%) in South Asia/ Central Asia/ East Asia and Pacific (Fig 5) and 44% (95% CI: 18–69%) ($I^2$ = 98.94%) in Latin America and the Caribbean (Fig 6). When studies selected based on HIV status were excluded, the results were similar for sub-Saharan Africa (39%; 95% CI: 26–52%;

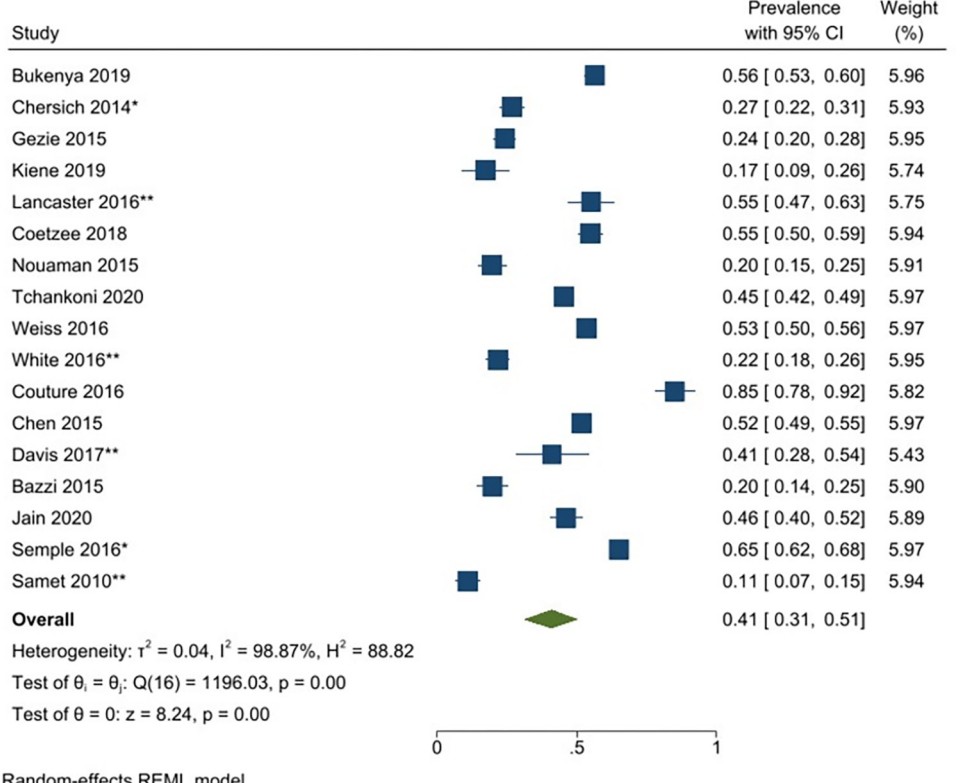

**Fig 2. Any hazardous/harmful/dependent alcohol use—pooled prevalence estimates.**

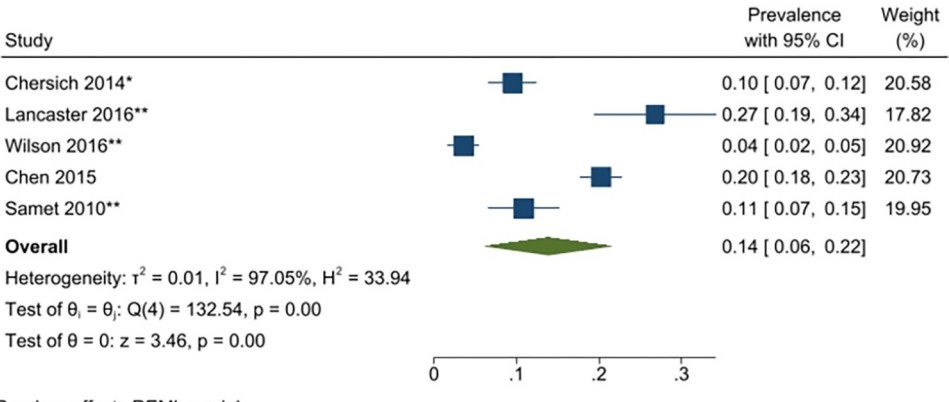

**Fig 3. Any harmful/dependent alcohol use—pooled prevalence estimates.**

n = 4) but differed for Asia (68%; 95% CI: 36–101%; n = 2) and Latin America (33%; 95% CI: 7–59%; n = 2).

## Daily alcohol use

Twelve studies reported data on daily alcohol use with a reported prevalence that ranged from 8–59% [29, 49, 71, 85, 87, 88, 90, 95, 96, 102, 130, 131]. The pooled prevalence of daily alcohol use among FSWs from LMICs is 26% (95% CI: 17–36%) ($I^2$ = 99.26) (Fig 7). Excluding one study that had selected participants based on HIV-status, the analysis yielded a similar estimated pooled prevalence of 24% (95% CI: 15–33%) ($I^2$ = 99.13). Prevalence estimates were also conducted for each of the different regions, resulting in an estimated pooled prevalence for daily alcohol use of 26% (95% CI: 11–41%) ($I^2$ = 99.20%) in Sub-Saharan Africa (Fig 8); 11.0% (95% CI: 8–15%) ($I^2$ = 89.16%) in South Asia/ Central Asia/ East Asia and Pacific (Fig 9)

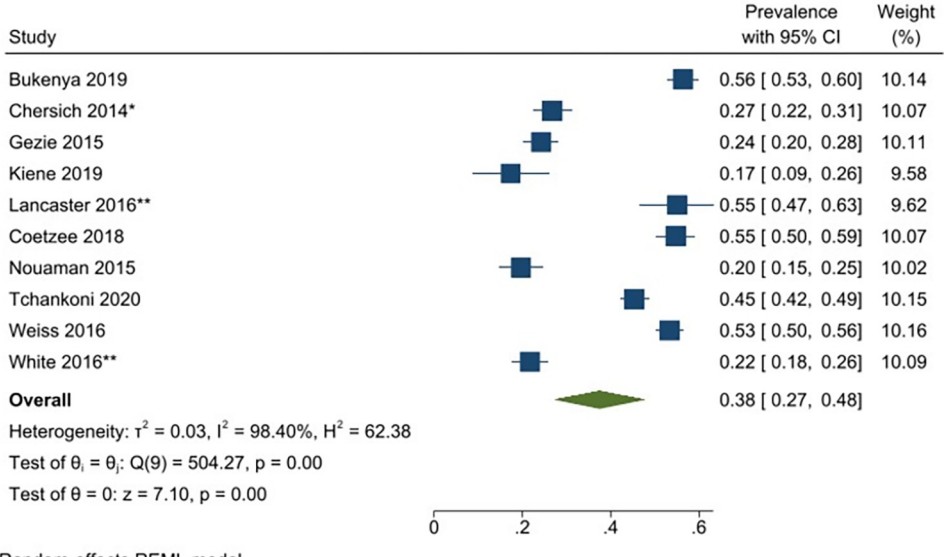

**Fig 4. Any hazardous/harmful/dependent alcohol use—pooled prevalence estimates for sub-Saharan Africa.**

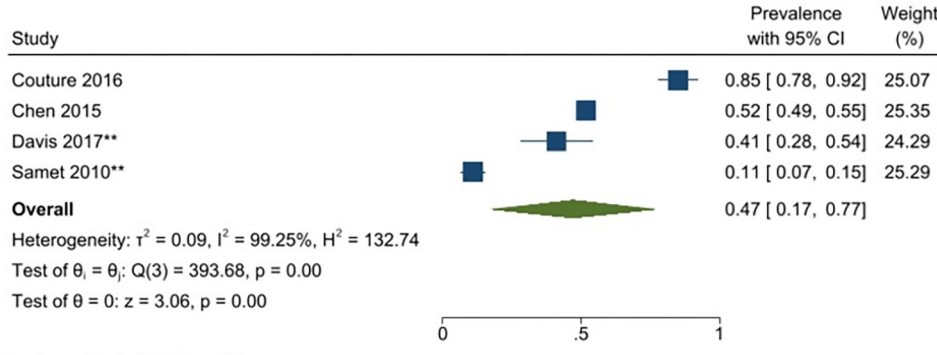

**Fig 5. Any hazardous/harmful/dependent alcohol use—pooled prevalence estimates for South Asia/ Central Asia/ East Asia and Pacific.**

and 37% (95% CI: 22–53%) ($I^2$ = 98.02%) in Latin America and the Caribbean (Fig 10). All pooled prevalence estimates are summarised in S5 Appendix.

## Associations between harmful alcohol use and other factors

We conducted subgroup meta-analyses to examine associations between harmful alcohol use and factors commonly experienced by FSWs (violence/police arrest, condom use, HIV/STIs, drug use and mental health problems). We only included studies that used a validated alcohol use measurement tool (Tables 2 and 4 –cross sectional studies and Tables 3 and 5 –cohort studies).

## Harmful alcohol use, violence and police arrest

In total, ten studies reported associations between violence and alcohol use [27, 28, 42, 50, 63, 84, 92, 96, 113, 118]. Of these, 3 studies used validated tools to measure alcohol use [27, 28, 113] (Table 2) and were included in the meta-analysis with a pooled unadjusted OR of 2.07; 95% CI: 0.63–6.76 (p-value = 0.23) (Table 4). Only one study reported on alcohol use and police arrest [28] (Table 2).

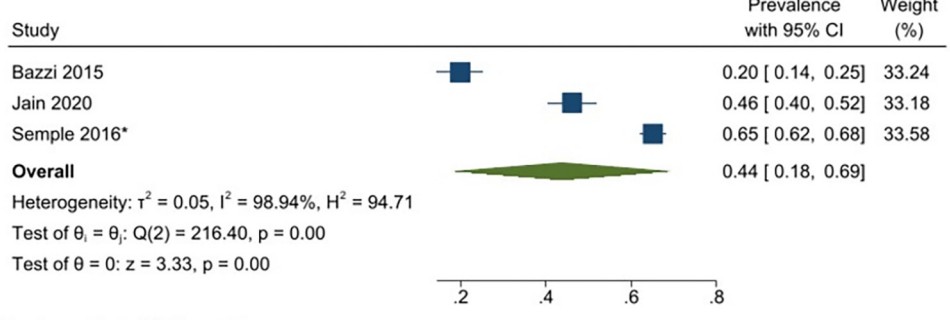

**Fig 6. Any hazardous/harmful/dependent alcohol use—pooled prevalence estimates for Latin America and the Caribbean.**

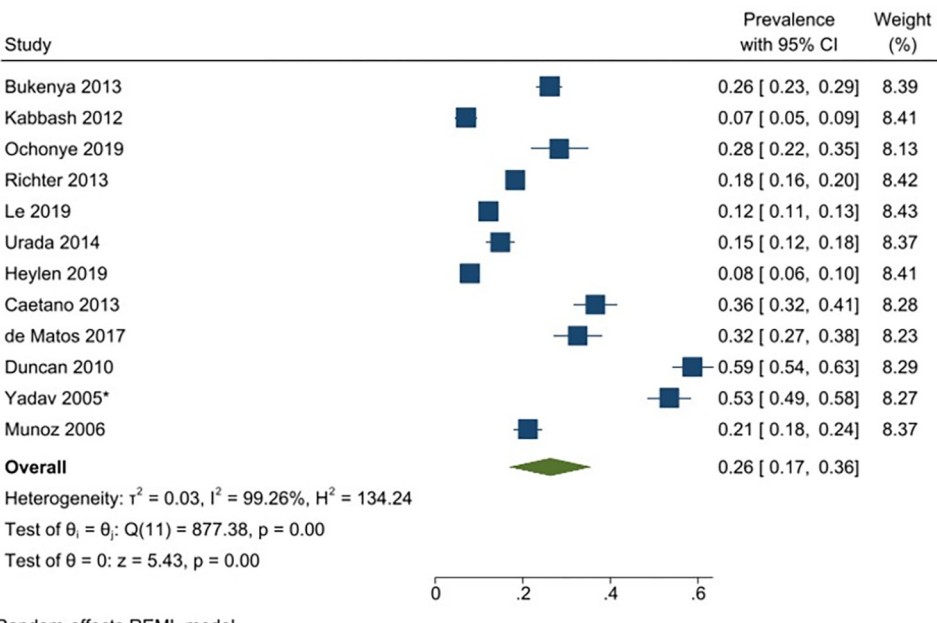

**Fig 7. Daily alcohol use—Pooled prevalence estimates.**

## Harmful alcohol use, condom use and other sexual risk behaviours

Seventeen studies [27, 28, 32, 33, 47, 52, 57, 59, 61, 63, 70, 80, 84, 89, 91, 102, 116] reported associations between condom use and alcohol use. Three studies using a validated alcohol use tool [28, 32, 33] included in a cross-sectional meta-analysis had a pooled unadjusted OR: of 1.31 (95%CI: 0.96–1.80; p-value = 0.09) (Table 4). Three studies [27, 55, 61] (Table 3) were included in a cohort meta-analysis with a pooled unadjusted RR of 1.65 (95%CI 1.01–2.67) for inconsistent condom use and alcohol use (excluding one study that had selected participants based on HIV-status: unadjusted RR: 1.66; 95%CI: 0.69–3.99) (Table 5).

Other associations reported between alcohol use and sexual risk behaviours included unplanned pregnancy [124], reduced use of any contraception [27], increased number of sexual partners [27, 28, 55, 80] and recent anal sex [63, 116]; however there were either

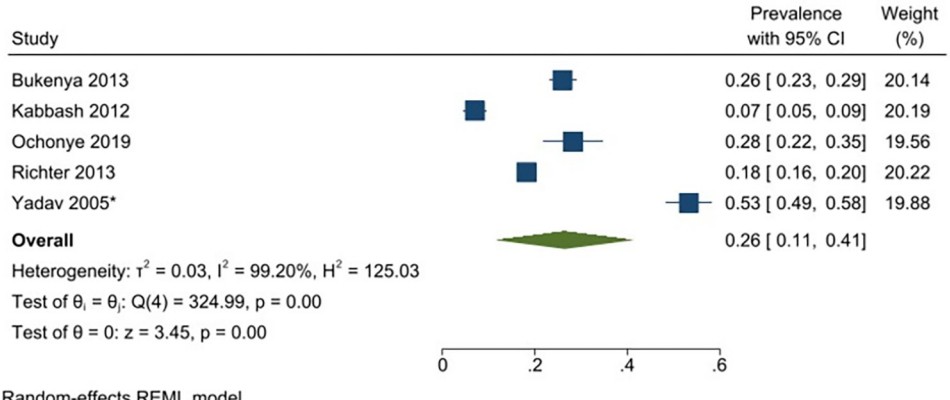

**Fig 8. Daily alcohol use—Pooled prevalence estimates for sub Saharan Africa.**

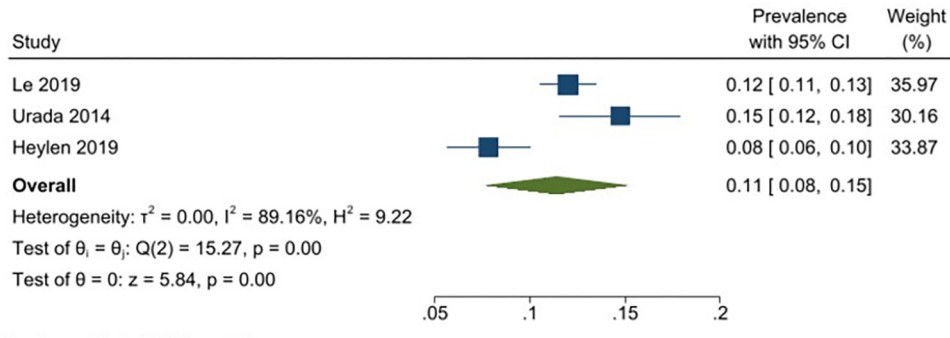

**Fig 9. Daily alcohol use—Pooled prevalence estimates for South Asia/ Central Asia/ East Asia and Pacific.**

insufficient studies to conduct a meta-analysis or validated measures were not used to measure alcohol use.

## Harmful alcohol use, STIs and HIV

Overall 12 studies [27, 32, 33, 60, 62, 63, 75, 78, 97, 101, 108, 113] reported associations between STI prevalence/incidence and alcohol use and one study reported associations with STI symptoms [100]. Three cross sectional studies [27, 33, 113] were included in meta-analysis (pooled unadjusted OR: 1.29; 95%CI 1.15–1.46); similar results were found when one study including only HIV negative FSWs was excluded (pooled unadjusted OR: 1.29; 95%CI: 1.14–1.44) (Table 4). Two cohort studies [55, 62] (Table 3) that reported on alcohol use and STI incidence were included in the meta-analysis, which indicated strong evidence of an association between harmful drinking and STI incidence (pooled unadjusted RR: 2.07; 95% CI: 1.40–3.05) (Table 5).

Nine studies [27, 33, 69, 97, 110, 126, 131] reported on associations between alcohol use and HIV. Meta-analysis among three studies using a validated alcohol use tool found no evidence of an association between harmful drinking and HIV prevalence [33, 73, 97, 126] (Table 4). One cohort study reported an association between hazardous alcohol use and HIV incidence (OR: 10.5; 95%CI: 1.27–87.58) (Table 3) [27].

Two studies reported associations between alcohol use and reduced ART adherence [54, 89], two reported on associations between alcohol use and viral suppression [54, 60] with one

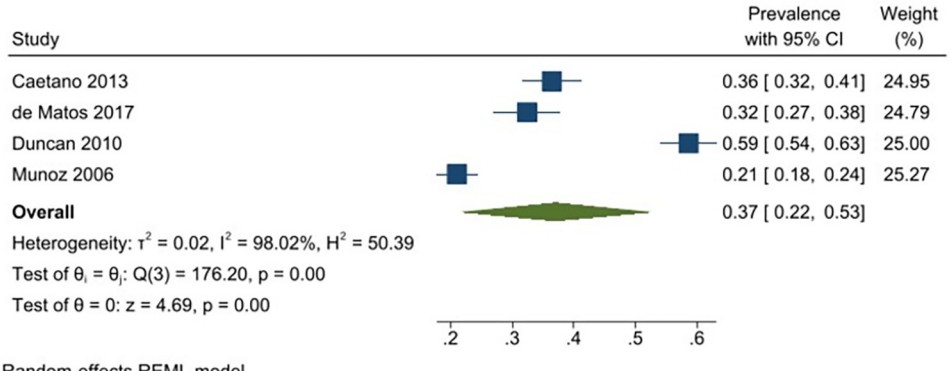

**Fig 10. Daily alcohol use—Pooled prevalence estimates for Latin America and the Caribbean.**

**Table 3. Associations with alcohol use (cohort studies).**

| Author & Study | Country | Sample | Alcohol use measure | Outcome of interest | | Sample size | Events* | Crude Rate ratio (95% CI) | P-value |
|---|---|---|---|---|---|---|---|---|---|
| Chersich (2014) | Kenya | HIV-negative FSWs | Hazardous or harmful drinking (AUDIT score > 8) | Unprotected sex with casual clients | | 399 | 90 | 2.15 (1.06–4.36) | 0.03 |
| | | | | Unprotected sex with regular clients | | 399 | 156 | 1.82 (1.20–2.76) | 0.005 |
| | | | | Unprotected sex with intimate partner | | 399 | 311 | 1.12 (0.96–1.31) | 0.2 |
| White (2016) | Kenya | HIV-positive FSWs | Hazardous or harmful AUDIT score 7–40 | Unprotected sex | | 405 | n/a | 2.59 (1.7–3.94) | <0.001 |
| Gezie (2015) | Ethiopia | FSWs | Problem drinking–answered yes to one of CAGE questions | Unprotected sex | | 467 | 297 | 1.06 (0.68, 1.68) | 0.8 |
| **HIV/STIs** | | | | | | | | | |
| Chersich (2014) | Kenya | HIV-negative FSWs | Hazardous or harmful drinking (AUDIT score >8) | HIV incidence | Low risk drinking | 399 | 2 | 2.82 (0.26–31.09) | 0.4 |
| | | | | | Hazardous drinking | 399 | 6 | 10.5 (1.27–87.58) | 0.03 |
| | | | | | Harmful/dependent drinking | 399 | 1 | 2.70 (0.17–43.29) | 0.5 |
| White (2016) | Kenya | HIV-positive FSWs | Hazardous or harmful AUDIT score 7–40 | STI incidence—Diagnosis of vaginal trichomoniasis, gonorrhea, or chlamydia at quarterly exams | | 405 | n/a | 2.03 (1.34–3.08) | 0.001 |
| Bazzi (2015) | Mexico | FSWs | AUDIT using cut-off >8 | STI incidence (chlamydia, gonorrhea, or active syphilis) | | 185 | n/a | 2.35 (0.75–7.36) | 0.1 |

*crude rates/no. of events were requested from authors where possible.

**Table 4. Associations with alcohol use (pooled OR).**

| Measure | Number of studies included | | Pooled unadjusted OR (95% CI) | P-value | studies |
|---|---|---|---|---|---|
| Any sexual and/or physical violence | | 3 | 2.07 (0.63–6.76) | 0.23 | Jain 2020 |
| | | | | | Semple 2016* |
| | | | | | Chersich 2014* |
| Inconsistent Condom use | | 3 | 1.31 (0.96–1.80) | 0.09 | Weiss 2016 |
| | | | | | Chen 2013 |
| | | | | | Semple 2016* |
| | Excluding studies with HIV negative FSWs only | 2 | 1.48 (0.96–2.27) | 0.08 | Weiss 2016 |
| | | | | | Chen 2013 |
| STI prevalence | | 4 | 1.29 (1.15–1.46) | <0.001 | Weiss 2016 |
| | | | | | Chersich 2014* |
| | | | | | Jain 2020 |
| | Excluding studies with HIV negative FSWs only | 2 | 1.29 (1.14–1.44) | <0.001 | Weiss 2016 |
| | | | | | Jain 2020 |
| HIV prevalence | | 3 | 0.46 (0.12–1.79) | 0.3 | Nouman 2015 |
| | | | | | Weiss 2016 |
| | | | | | Couture 2016 |
| Depression | | 2 | 1.1 (0.52–2.30) | 0.8 | Jain 2020 |
| | | | | | Coetzee 2018 |
| Illicit Drug use | | 3 | 2.44 (1.24–4.80) | 0.04 | Jain 2020 |
| | | | | | Chersich 2014* |
| | | | | | Semple 2016 |
| | Excluding studies with HIV negative FSWs only | 2 | 1.94 (0.90–4.19) | 0.1 | Jain 2020 |
| | | | | | Semple 2016 |

**Table 5. Associations with alcohol use (pooled RR).**

| Measure | Number of studies included | | Pooled unadjusted RR (95% CI) | P-value | studies |
|---|---|---|---|---|---|
| Inconsistent Condom use | | 3 | 1.65 (1.01–2.67) | 0.04 | White 2016 |
| | | | | | Gezie 2015 |
| | | | | | Chersich 2014* |
| | Excluding studies with HIV negative FSWs only | 2 | 1.66 (0.69–3.99) | 0.3 | White 2016 |
| | | | | | Gezie 2015 |
| STI incidence | | 2 | 2.07 (1.40–3.05) | 0.0003 | White 2015 |
| | | | | | Bazzi 2015 |

reporting an association between harmful alcohol use and viral non-suppression [54]. Due to differences in how these studies measured alcohol and ART adherence/viral suppression, pooled OR were not calculated. One study reported an association between alcohol use and HIV status awareness [53]. There were also associations reported between alcohol use and Hepatitis B [76, 88] and Hepatitis C infection [77, 130]. Two studies reported an association between alcohol use and PrEP use with one finding no association between alcohol use and perceived barriers to oral-PrEP use (based on statements across five domains about why they would not use PrEP) [113] and another finding an association between alcohol use and reduced oral-PrEP adherence [52].

## Harmful alcohol use, depression and illicit drug use

Overall, eight studies [83, 86, 104, 105, 107, 113, 123, 128] reported associations between alcohol use and mental health problems. Two studies [107, 113] that reported on associations with depression were included in a meta-analysis with a pooled unadjusted OR of 1.1 (95%CI: 0.52–2.30). A total of six studies [27, 28, 74, 79, 113, 119] reported on associations between alcohol and any illicit drug use, of which three were included in a meta-analysis with a pooled unadjusted OR of 2.44 (95%CI 1.24–4.80) [27, 28, 113]. When one study including only HIV-negative FSWs was excluded the pooled unadjusted OR was 1.94 (95%CI: 0.90–4.19).

## Alcohol use interventions

Of the four studies reporting on alcohol use interventions, 3 were in sub-Saharan Africa (2 reported on the same study) and 1 was in East Asia and the Pacific. An RCT in Kenya assessed an intervention involving 6 counselling sessions based on the WHO Brief Intervention for Alcohol Use. It reported a statistically significant reduction in alcohol use and binge drinking in the intervention group as well as reductions in violence from clients [39, 40]. An intervention in South Africa [43] assessed an empowerment-based two-session HIV intervention designed to reduce sexual risk, substance use, and violence victimization among at-risk women. At 6 months it found women who received the intervention reported a significantly lower mean numbers of days drinking alcohol in the previous 30 days, were less likely to meet DSM-IV criteria for alcohol dependence, were more likely to report using a condom at last sex with a main partner, and were less likely to report sexual abuse by a main partner in the previous 90 days. An RCT in Mongolia [48] found a reduction in AUDIT score before and after a range of interventions including motivational interviewing, wellness promotion and a relationship-based HIV sexual risk reduction intervention.

## Discussion

In this systematic review and meta-analysis using data from 87 unique studies and including 51,904 FSWs from 32 LMIC countries across all global regions, we found a high prevalence of daily and harmful alcohol use among FSWs associated with a range of risk factors. According to our pooled prevalence estimates two-fifths (41% (95% CI: 31–51%)) of FSWs reported any hazardous/harmful/dependent alcohol use and one quarter (26% (95% CI: 17–36%)) reported daily alcohol use. The global prevalence of alcohol use disorders among women in the general population is 5.1% [132] indicating a significantly higher burden of harmful alcohol use among FSWs. Alcohol use is prevalent during sex work and on entry into sex work, which reflects previous evidence about the availability and normalisation of alcohol in the sex work industry [13, 14]. The high burden of alcohol use has serious health and social implications for FSWs, as excess alcohol use is associated with multiple poor physical and mental health outcomes [133]. Meta-analyses found significant associations between problem alcohol use and inconsistent condom use, increased STI prevalence and incidence and other drug use among FSWs. The associations between harmful alcohol use and wider social and occupational risk factors as well as the high levels of alcohol use during sex work indicate the need to tackle upstream and structural risk factors in future interventions.

Societal standards and norms greatly influence alcohol and drinking patterns. Research from LMICs suggests that policies overseeing the availability of alcohol and legal drinking age are steady predictors of alcohol consumption [134]. Alcohol is widely available in the sex work industry [13] and findings from this review support the fact that alcohol use during sex work is highly prevalent. This correlates with findings from qualitative studies, which report sex workers use alcohol as a way of coping with the challenges of work [14] and due to pressure from clients [135, 136]. In addition, substance use other than alcohol is common during sex work [13, 22, 92], and we found other drug use was associated with harmful alcohol use. Future interventions should consider addressing poly-substance use and tackling social norms around alcohol and other drug use in the sex work environment.

Research from LMICs indicates that alcohol industry marketing has focussed advertising on increasing uptake of drinking among young women [5, 137]. Alongside our findings of a high prevalence of alcohol use on entry into sex work, this indicates that young FSWs should be a target for alcohol interventions to prevent long term alcohol related harms. On a structural level, advocating for wider policy changes around alcohol pricing, accessibility and advertising is also important as these factors have been shown to be crucial in tackling consumption globally [5, 138–140]. In addition, advocating for the de-criminalisation of sex work [12] in the majority of countries globally would make it easier to regulate and improve the safety of the sex work environment.

We did not find associations between alcohol use and violence, however the number of studies included in meta-analysis (n = 3) were limited. Previous associations have been found between alcohol use and intimate partner physical or sexual violence victimization among women in the general population [141]. Interventions addressing violence have been shown to be effective among women in LMICs [142, 143] as well as among FSWs [144], and these should be integrated into future alcohol use interventions. We did not find an association between alcohol use and HIV although this may be due to the limited number of studies reporting alcohol use with a validated tool (n = 3), and in particular lack of longitudinal studies (n = 1), in our meta-analyses. Previous systematic reviews in the general population have reported longitudinal associations between alcohol use and HIV infection [145, 146] and between alcohol use disorders and decreased adherence to antiretroviral therapy and poor HIV treatment outcomes among people living with HIV [147]. Previous quantitative and qualitative research has

also indicated that alcohol use may be a barrier to oral-PrEP use [148, 149] including among FSWs [52, 82]. However, the evidence is mixed with other studies reporting no association [113]. Longitudinal studies that measure oral-PrEP adherence, rather than just self-reported oral-PrEP acceptability, are needed to better understand this potential association. We found associations between alcohol use and other sexual risk behaviours including reduced condom use and increased STI infection prevalence and incidence, which echoes findings among women in the general population [150, 151]. Given the well-established links between condom use, STI infection [152] and HIV risk, our findings suggests that alcohol use interventions should be embedded within existing HIV/STI and sexual health services for FSWs, with the view to providing more wholistic integrated services that address women's physical, psychological and social well-being.

The high levels of alcohol use among FSWs, compared to the general population, and associated risk factors can be considered through a syndemics framework. Syndemic theory aims to identify how the combined effects of health or social epidemics in a population, such as harmful alcohol use and violence, exceeds the sum of their independent components [25, 26] and attempts to identify those most at need as a result of syndemic risks in order to deliver comprehensive and targeted health and social services. To date, the identification of syndemics has largely relied on the use of quantitative methods, which do not go beyond the simple summing of the total number of conditions in individuals [153]. Two previous studies, to our knowledge, among FSWs have examined substance use, violence and HIV from a syndemics perspective, both in relation to oral-PrEP uptake [82, 113]. Only four studies reported on alcohol use interventions for FSWs, all of which are focussed on addressing alcohol use at the individual level only. The syndemic interactions between alcohol use and occupational and socioeconomic risk factors associated with sex work are crucial to understand for policy makers developing alcohol use interventions for FSWs, and this should be a key area for future research.

## Limitations

We conducted a comprehensive literature search in line with the PRISMA guidelines, with independent screening and quality appraisal of all studies. Our review captures a broad range of studies from across a variety of geographic regions. Despite this, our review had limitations. Our search was limited to published studies and those written in English and hence we may have excluded important studies. There is a risk of publication bias particularly when interpreting the pooled OR for association, as studies not reporting significant association may be less likely to be published. In addition certain geographic regions are over-represented, such as sub-Saharan Africa, meaning that results may not be generalizable to all settings. We report on unadjusted ORs for the association between harmful alcohol use and potential risk factors, which allows for direct comparisons between studies; however there is a risk that this may have led to over or under-estimated associations. When examining associations, we only included studies in meta-analyses that used a validated alcohol use tool. However, the measures used for associated risk factors such as violence were not all based on validated tools, which could have led to information bias. We included studies in which participants were sampled based on a potential risk factor for alcohol use such as HIV status; however we ran seperate analyses which included and excluded studies that sampled based on HIV status (mainly HIV negative FSWs). We think it is important to show both estimates given the small number of studies eligible for meta-analysis. There was a lack of longitudinal studies (n = 6). Longitudinal studies are necessary to understand the direction of association between alcohol use and common health and social concerns. FSWs are considered a 'hard to reach population' due to

the stigmatised and illegal status of sex work. As a result FSWs are an inherently difficult population to sample and in many settings, probability sampling is not possible. Due to the sampling strategy of the majority of studies (non-probability sampling) there is a risk that the most vulnerable women were excluded, which may have led to lower estimates of alcohol use. This is a well-established concern in the FSW literature [154] and must be taken into account when assessing study quality, as non-probability sampling techniques such as snow-ball and convenience sampling can lead to selection bias. Techniques such as respondent driven sampling (RDS) that uses a chain referral methodology to collect data from hard-to-reach populations such as FSWs, have been developed to reduce the biases found in standard snowball sampling methods [154, 155]. Future research among FSWs should aim to use sampling methods such as RDS in order to reach the most hidden members of the FSWs population and limit selection bias. Another key limitation was the variability in measurement tools for alcohol use between studies, with many studies not using a validated tool. Even where the same tools were used such as AUDIT, cut-off scores were not always uniformly applied. As a result, comparability and reliability of findings between studies is limited. The use of standardised measurement tools, such as those recommended in the STRIVE 'Measuring alcohol-related HIV risk' technical briefing [156], and longitudinal study designs should be prioritised in future research on alcohol use among FSWs.

## Conclusions

This is the first systematic review, to our knowledge, to estimate the prevalence of harmful and daily alcohol use among FSWs in LMICs and to report on associations with common health and social concerns. Our findings suggest that FSWs experience a high burden of daily and problem alcohol use and that harmful drinking is associated with a number of syndemic risk factors including inconsistent condom use, STIs and drug use. There were few alcohol use interventions described; all of these focussed on individual level behaviour change rather than the wider sex work environment that encourages women to engage in excess alcohol consumption. Future research should seek to better understand the syndemic nature of risks, which contribute to high levels of harmful alcohol use among FSWs, and develop wholistic multilevel interventions that address not just individual-level but societal and structural risks such as gender inequality, stigma and poverty.

## Supporting information

**S1 Appendix. Search strategies.**
(DOCX)

**S2 Appendix. Centre for evidence based medicine critical appraisal tool.**
(PDF)

**S3 Appendix. Quality assessment of studies.**
(DOCX)

**S4 Appendix. List of Stata commands.**
(DOCX)

**S5 Appendix. Table of pooled prevalences.**
(DOCX)

**S6 Appendix. PRISMA 2020 abstract checklist.**
(DOCX)

**S7 Appendix. PRISMA 2020 checklist.**
(DOCX)

## Author Contributions

**Conceptualization:** Tara S. Beattie.

**Data curation:** Alicja Beksinska, Oda Karlsen.

**Formal analysis:** Alicja Beksinska.

**Funding acquisition:** Tara S. Beattie.

**Investigation:** Alicja Beksinska, Oda Karlsen.

**Methodology:** Alicja Beksinska, Oda Karlsen.

**Supervision:** Mitzy Gafos, Tara S. Beattie.

**Validation:** Oda Karlsen.

**Writing – original draft:** Alicja Beksinska, Oda Karlsen.

**Writing – review & editing:** Mitzy Gafos, Tara S. Beattie.

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
