## [Decision Letter · Decision Letter 0]

3 Aug 2022

PGPH-D-22-01100

Alcohol use among female sex workers in low-and middle income countries: A systematic review and meta-analysis

Dear Dr. Beksinska,

Thank you for submitting your manuscript to PLOS Global Public Health. After careful consideration, we feel that it has merit but does not fully meet PLOS Global Public Health’s publication criteria as it currently stands. Therefore, we invite you to submit a revised version of the manuscript that addresses the points raised during the review process.

Please address all the comments raised by the reviewers. 

We look forward to receiving your revised manuscript.

Kind regards,

Palash Chandra Banik, MPhil

Academic Editor

Journal Requirements:

1. Please provide separate figure files in .tif or .eps format and ensure that all files are under our size limit of 10MB.

2. Please include your main tables as part of your main manuscript and remove the individual files.

3. Please ensure that you refer to Figure 10 in your text as, if accepted, production will need this reference to link the reader to the figure.

4. We noticed that you used “unpublished” in the manuscript. We do not allow these references, as the PLOS data access policy requires that all data be either published with the manuscript or made available in a publicly accessible database. Please amend the supplementary material to include the referenced data or remove the references.

5. We have noticed that you have uploaded Supporting Information files, but you have not included a list of legends. Please add a full list of legends for your Supporting Information files after the references list.

Additional Editor Comments (if provided):

Reviewers' comments:

Reviewer's Responses to Questions

**Comments to the Author**

1. Does this manuscript meet PLOS Global Public Health’s publication criteria? Is the manuscript technically sound, and do the data support the conclusions? The manuscript must describe methodologically and ethically rigorous research with conclusions that are appropriately drawn based on the data presented.

Reviewer #1: Yes

Reviewer #2: Yes

Reviewer #3: Yes

Reviewer #4: Yes

Reviewer #5: Yes

Reviewer #6: Yes

Reviewer #7: Yes

2. Has the statistical analysis been performed appropriately and rigorously?

Reviewer #1: Yes

Reviewer #2: Yes

Reviewer #3: Yes

Reviewer #4: I don't know

Reviewer #5: Yes

Reviewer #6: Yes

Reviewer #7: Yes

3. Have the authors made all data underlying the findings in their manuscript fully available (please refer to the Data Availability Statement at the start of the manuscript PDF file)?

Reviewer #1: Yes

Reviewer #2: Yes

Reviewer #3: Yes

Reviewer #4: Yes

Reviewer #5: Yes

Reviewer #6: Yes

Reviewer #7: Yes

4. Is the manuscript presented in an intelligible fashion and written in standard English?

Reviewer #1: Yes

Reviewer #2: Yes

Reviewer #3: Yes

Reviewer #4: Yes

Reviewer #5: Yes

Reviewer #6: Yes

Reviewer #7: Yes

5. Review Comments to the Author

Reviewer #1: Manuscript review: PGPH-D-22-01100

Overall assessment

This is a promising and important study about a relevant and current topic in global public health. Overall, the study is methodologically sound, adheres to the current standards of a modern epidemiological study, and merits publication, but some sections, especially methodology and results, are too brief, and need to be expanded and more detailed. The language of the study appears to reflect that the main writer(s) are not native speakers of English and are new to scientific writing. There are minor misconceptions regarding study designs and the measures of disease frequency and effect they produce. The quality of the writing can be improved, as the manuscript in its present form does not reflect the quality and amount of work that was put in this research project. The senior author, especially if he/she is a primary speaker of English, should review and edit carefully. The junior authors of the study seem to need guidance and the manuscript will benefit greatly from feedback from a more senior researcher, which can pick up on several areas for improvement which this reviewer noticed, but which are not the task of a reviewer to correct. This study would also have benefitted from a narrower aim. The decision to include in the same systematic review the aim to estimate prevalence of problem alcohol use among FSWs and the association of alcohol use disorder with other exposures experienced by FSWs (HIV, STIs, violence, etc.), made this systematic review and the manuscript unnecessarily more complex, especially because the kind of study designs that answers those questions are distinct. Cross-sectional design is the ideal design to estimate prevalence, while case-control studies do not allow to calculate prevalence, since we don’t have access to the source population that produced the cases. This reviewer questions the decision to include experimental studies, despite the authors attempt to justify that they were including experimental studies with baseline assessments. An experimental study is an artificial situation that is designed to answer a particular question, and often involves populations and contexts that make those studies feasible. This reviewer would argue that any experimental study included provides a biased estimate of prevalence of alcohol use among FSWs, given the inherent nature of experimental studies, which have strict inclusion criteria and willingness of participants to receive intervention being proposed by the investigators. Given the potential of selection bias inherent to any experimental studies, this reviewer would argue against the inclusion of any experimental design in this present study. In retrospect, the authors probably should have concentrated in providing an estimate for the prevalence of the distinct forms of alcohol use, while studying the association of alcohol use with other exposures/outcomes of interest to a separate study. The systematic review of the association of alcohol use with other risk factors lived by FSW should have been saved for a separate study. In trying to include everything in the same study, several methodological aspects of study design and results did not get the space they deserved. One area of particular concern, and which should have received great deal of attention, is a comprehensive and thoughtful discussion about the potential for bias, especially selection bias, in each original studies included in the meta-analysis. Despite the authors argument that they used a tool to critically appraise the original studies, the report that 27 studies scored as high quality, 62 scored as moderate and 9 scored as weak, is surprising. Just the fact that most of the original studies did not use probabilistic sampling already makes them at very high risk of selection bias, and that on itself would be enough to make the evidence provided by a study weak, at best. The critical appraisal tool is a basic tool to screen studies for inclusion in the systematic review, but should not serve as the only measure of the potential for bias in the estimates used in the meta-analysis. Besides, item 5 of the appraisal tool “Was the sample of subjects representative with regard to the population to which the findings will be referred?” was answered “yes” for almost all the studies. That is not correct, when item 4 “Could the way the sample was obtained introduce (selection)bias?” was also answered “yes” for nearly all the studies. If item 4 is a “yes”, then item 5 is automatically a “no”, or “unknow”, at best. If there is potential for selection bias, we can’t know if the study sample is representative of the target population; and with the use of non-probability sampling, that is almost certainly a “no” on item 5 for nearly all the studies. Nearly all the original studies included in this meta-analysis are at great risk of bias and the authors do not do enough to address that fundamental issue with this meta-analysis. This fundamental risk of bias in the pooled estimate, originating from the individual studies, needs to be addressed in the introduction, methods, results, and discussion. The authors touch on that, but it requires a more comprehensive treatment. This writer points the authors to the PRISMA statements items (below) that require a more comprehensive approach to the risk of bias in the original studies.

Lastly, any meta-analysis needs to address the risk of publication bias, especially in a topic of studies in LMICs. The authors did not address sufficiently the risk of publication bias and that should be included in the methods, results, and discussion section, including producing and reporting a funnel plot. Similarly, this writer points the authors to the PRISMA statement items that require a more comprehensive treatment of the risk of publication bias in a systematic review and meta-analysis.

Specific comments

TITLE

“low-and middle income countries”

Correct way to write is “low- and middle-income countries”.

INTRODUCTION

“In recent years alcohol use has been increasing in many LMICs.”

Needs reference.

“which corresponds to the AUDIT tool and some other alcohol use tools.”

Not professional English; “some other” is not a professional and scientific statement in this case.

METHODS

Methods section needs to be expanded and more detailed. In particular the following items on the PRISMA statement:

• 11 Specify the methods used to assess risk of bias in the included studies, including details of the tool(s) used, how many reviewers assessed each study and whether they worked independently, and if applicable, details of automation tools used in the process.,

• 14 Describe any methods used to assess risk of bias due to missing results in a synthesis (arising from reporting biases), and

• 15 Describe any methods used to assess certainty (or confidence) in the body of evidence for an outcome.

“This review included studies which reported any measure of prevalence or incidence of alcohol use on the basis of a clinical interview, self-reported or clinical examinations among FSWs even if sex workers were not the main focus of the study.”

“The following study designs were included: cross-sectional survey, case–control

study, cohort study, case series analysis, or experimental study with baseline

measures for alcohol use.”

Case-control studies, case series, and clinical trials do not allow for calculation of prevalence, nor incidence (theoretically you can estimate incidence in a clinical trial, but not with the outcome of interest in this study). The authors need to clarify the inclusion criteria or explain the contradiction that they intended to include only studies with measure of prevalence or incidence of alcohol use disorder and yet list case-control, case series, or an experimental study design. While including case-control studies or cohort studies would have helped estimate strength of association of alcohol use with violence, STIs, inconsistent condom use, etc, those studies would not meet the including criteria of reporting prevalence or incidence of alcohol

“Centre for Evidence-Based Management (CEBM) Critical Appraisal for Cross-Sectional Surveys Tool”

They should name and abbreviate correctly as Center (notice the American spelling on the name on supplement 2), and abbreviate as CEBMa, otherwise it is confused with the Centre for Evidence-Based Medicine (CEBM) at Oxford (https://www.cebm.ox.ac.uk/resources/ebm-tools/critical-appraisal-tools).

“Through narrative synthesis of available measures and known validated tools we selected a range of measures of alcohol use and harmful alcohol use for meta-analysis.”

Can they clarify? What do they mean?

They have to describe how they extracted information on association of alcohol use with other outcomes. This needs to be greatly expanded in the methods section and the discussion along the lines of the PRISMA items pointed out above.

The authors should mention which Stata 16 command they use to perform meta-analysis of proportion and odds-ratio.

RESULTS

The results need to be improved concerning the following PRISMA statement items: 18 (Present assessments of risk of bias for each included study.), 21 (Present assessments of risk of bias for each included study.), and 22 (Present assessments of certainty (or confidence) in the body of evidence for each outcome assessed.)

“The initial electronic search yielded 416 results, with 19 additional studies identified through reference list screening and online searches.”

The methods section did not say anything about screening reference list and on-line search. That is not a “systematic review” approach, and those methods should not have been used, since another investigator would not be able to reproduce the same search. “Augmenting” the systematic review by combing through references or the internet defeats the purpose of a systematic review, which is to produce a comprehensive, reproducible literature search.

Table 1, column 1: did they mean author and year?

Table 1, column 4 “Sampling procedure”, the term non-random sampling is not descriptive; did they mean non-probabilistic or non-probability sampling? That is a more appropriate term than “non-random sampling”. What do they mean by non-random? All the other methods of sampling listed (venue-based, convenience, snowball, respondent-driven, purposive, targeted, etc.) are non-probabilistic/non-probability (“non-random”) sampling techniques. Please clarify what they mean by “non-random sampling” and why they chose to use that term for some studies and not for others, since nearly all the studies used non-probability sampling. I am guessing the authors of the original article did not describe the sampling procedure used.

Also, there is a heterogeneous use of terms. All the sampling techniques described, snowball, respondent-driven, target, purposive, venue-based, etc, despite their fancy names, amount to non-probability sampling, also referred to as convenience sampling.

The authors should consider categorizing the type of sampling used in each study as either probability sampling, and then describe the type of probability sampling technique used (simple random, stratified, cluster, multistage, systematic, complex), or non-probability sampling, and then list the type used.

“A variety of measures on drinking prevalence and frequency were reported with the

majority of studies not using a validated tool to assess alcohol use.”

This sentence is too vague. If this is being presented in results, stating “the majority” is not sufficient; how many studies and what percentage of studies did not use a validated tool? For example, of the N studies included in this meta-analysis, n (%) used a validated tool to assess alcohol consumption, while n (%) did not.

Table 1, Column 6, Method of assessing outcome

Fawole (2004) “questionnaire”, this is not sufficiently descriptive.

Leddy (2018) “survey”

Ochonye (2019) “Semi-structured interviewer administered

Pattern of alcohol use

This subsection of the results section reporting of the patterns of alcohol seems redundant. The authors set out to study hazardous, harmful, and alcohol dependence and should have concentrated on the defined outcomes for this study. Yet they provide a description of using alcohol while engaging in sex work.

Table 2 seems redundant, since result of meta-analysis is already presented as a forest plot, and increases the size of publication. The authors should consider if the information in table 2 is not already present in the forest plots and other parts of the results.

Harmful alcohol use

“A meta-analysis was conducted with seventeen unique studies that used a validated tool (excluding studies which included only substance using FSWs) and estimated the pooled prevalence of any harmful alcohol use to be 41% (31-51%).”

If the numbers in parenthesis are the confidence interval, the author should have written (95% CI, 31% to 51%), or indicated that it is the range of values; just writing the limits of the confidence interval in the narrative portion is not the standard for reporting, otherwise the reader does not know what is being reported between parenthesis.

The same comment goes for all the numbers in parenthesis in the section “Harmful alcohol use”.

One important portion of the results section is that the authors need to provide a funnel plot to show that there was no publication bias. This can be accomplished with meta funnelplot command in Stata.

Alcohol use interventions

Unclear why there is a section on alcohol use interventions. At any point during the introduction and methods the authors indicate that this study was interested in reporting on alcohol use interventions.

DISCUSSION

The main focus of this study, prevalence of alcohol use among female sex workers, is not a simple outcome to ascertain. The same can be said about the other outcomes (STIs, HIV/AIDS, violence, etc.). Estimates obtained in the original studies are at high risk for selection and information bias, and confounding. Although the investigators did employ a tool to rate the quality of the study, special attention and space in the article should be dedicated to discussing the risk of bias in the original estimates.

Although the authors do mention the fact that most studies used non-probability sampling methods in the discussion section, they need to greatly expand on that portion of discussion since this is the foundation of the validity of all the work put in this study.

They need to discuss the non-probabilistic sampling procedures used in this “hard-to-reach” population and implications to bias and accuracy of estimates of prevalence.

The authors also need to provide an assessment about the risk of bias and confounding for the association of alcohol use with other exposures/outcomes in the original studies.

A discussion about publication bias needs to also be included.

STYLE

Manuscript should have been submitted double-spaced.

Reference should be in brackets; lines should be numbered.

Formatting of paragraphs need to be fixed. Too many paragraphs are making it difficult to read.

The authors don’t need to reference each study included in the meta-analysis when reporting the results in the body of the text; some statements in the result section are followed by a long list of references. That is not necessary. If the information is relevant it should be in the table describing the studies and the narrative portion of results should make reference to the table.

Not sure why the editorial staff accepted this manuscript that does not follow the basic rules of submission to PLOS GPH.

Reviewer #2: The authors conducted a systematic review and meta-analysis of alcohol use among female sex workers in LMICs. Their work is well-written and results are presented clearly.

I have a few comments that should be addressed before publication of this work.

(1) The authors used the Appraisal tool for Cross-Sectional Studies to classify study quality. Please list a few quality indicators to give the reader an idea on what criteria are important in this quality characterization. The classification thresholds (0–4,5–7,8–11) were chosen by the authors not by CEBM, right? This should be clarified too.

(2) In the study quality scoring “authors compared 10% of the results of scoring and discussed disagreements in scoring to ensure uniformity in the quality assessment process.” Can the authors please comment on score differences? Some of the used criteria may not be completely objective and it would be of interest to report score differences.

(3) Did the authors use “study quality” to weigh the importance of certain results w.r.t. others? It didn’t seem like that based on the described results, but it would be helpful for the reader to clearly state the reason for reporting “study quality” scores. I guess that the authors just intended to provide some results on study quality distributions in this specific field?

(4) A brief overview of the Higgins statistic would be helpful for the reader. How should small and large values of l^2 be interpreted? I also recommend that the authors add the corresponding reference in their manuscript (https://www.bmj.com/content/327/7414/557).

Minor issues:

“weak quality .” → “weak quality.”

“Figure 1.” → “Figure 1:” (inconsistent use of “:” and “.” in figure legends)

Reviewer #3: Overall, this study summarized the results of 22 years of research well and drew good meta-analysis results.

Only minor details need to be corrected.

1. There is no need to indicate statistically non-significant results in the abstract

"Harmful alcohol use was significantly associated with inconsistent condom use (pooled unadjusted OR:1.31; 95%CI: 0.96-1.80)"

It is better to use RR 1.65 (1.01-2.67) among the contents of Table 4b for this content.

2. All figure numberings from "Harmful alcohol use" on page 12 are incorrect. should be corrected

Figure 1 -> Figure 2

Figure 2 -> Figure 3

Figure 3 -> Figure 4

Figure 4 -> Figure 5

Figure 5 -> Figure 6

Figure 6 -> Figure 7

Figure 7 -> Figure 8

Figure 8 -> Figure 9

Figure 9 -> Figure 10

Reviewer #4: This study is good and brings up important aspects on this key population. However, in the abstract, the authors mentioned that there is 'no quantitative synthesis', I would suggest they change that statement, to say 'to our knowledge' as the quantitative synthesis could exist elsewhere (grey literature etc.) but they are not aware of it.

Secondly, they mentioned that there was an association between alcohol use and inconsistent condom use (OR 1.31 (0.96 - 1.80). The CI passes through 1 and therefore the association is not significant.

There is a typing error where they said Tabe 1 rather than Table 1 under sample techniques.

Reviewer #5: Comments for authors

This study was very well researched and written and clearly outlines the methodology for inclusion, quality of studies included, as well as the major findings. The study is clearly an important one. A major conclusion, about the fact that interventions are usually targeted at the individual level, rather than the environmental level, was noted in the Conclusion. Perhaps just a few more lines could be added about this very important finding, with perhaps a few lines added in the Introduction to discuss the syndemic nature of this phenomenon.

General comments

Overall well-written, though several times authors used “which” instead of “that” – a quick read to correct would be useful.

Specific comments

Abstract

In background, would have been helpful for authors to note why this topic is important. Overall, clearly written and informative.

Introduction

An important point made in the introduction – that since high income countries are implementing a range of rules/restrictions on alcohol, hence the need for alcohol companies to find replacement users – would have been impactful if noted in the Abstract.

In the first paragraph, authors note the use pattern of alcohol use in sub-Saharan Africa. Would have been important to also include a comparison of what the use pattern is in high income countries.

Methods

Inclusion criteria – authors note they included studies that included FSW as participants, even if FSW were not the main focus of the study. How did they parse out the findings specifically for the FSW? Or did authors only include studies where the original authors already separated out findings for FSW?

- Data extraction

Line 6 - Meta-analysis should be changed to meta-analyses

Results

- Harmful alcohol use and STIs

Third paragraph – authors note that several studies reported on associations between alcohol use and ART adherence and viral suppression, however it is not clear what the direction of association was, i.e., positive or negative association. The associations were more clearly explained with PrEP use.

Discussion

In the Results, authors noted that there was very high use of alcohol during first sex work. Would have been interesting, though not necessary, to add a bit of detail about this in the Discussion.

Reviewer #6: This paper discusses the prevalence of harmful alcohol use among FSWs in LMICs and its associations with health and social concerns like violence/police arrest, condom use, HIV/STIs, drug use and mental health problems.

Minor comments:

1. The title of the paper does not accurately reflect the topic covered.

2. Have you thought about including grey literature?

Reviewer #7: Research Topic : Please remove the hyphen between "low" and "and"

Abstract:

Background. How should the first statement be written? I guess it should be "low and middle income countries" Please double check and kindly revise. Methods: case series analysis and experimental studies were captured only in the abstract and never mentioned again. Kindly check and make the necessary changes. Conclusion: your inclusion criteria means you did not consider longitudinal studies. You can therefore not make a conclusion that 'there is limited longitudinal studies based on your study.

Introduction: Do not use this "()" to enclose the intext reference. Please use this instead "[]". it makes recognition of intext reference difficult.

Paragraph 2: Please cite a reference for the sentence beginning with 'Sociocultural and economic factors...........

Paragraph 4 Line 1 Please reconsider reframing the sentence ...... but there is generally............

Methods:

Inclusion criteria: Was the year of publication of the selected papers included. please indicate it under this section

Quality assessment: Elaborate on how the scoring was done. How did you score "yes", "No" and "cannot tell" and how did you arrive at the overall score for each article.

NB: None of the appendices is labelled A. I guess you mean appendix 1 but appendix 1 does not contain the information you are referring readers to. its rather the 3rd appendix. Kindly check and correct.

Results: How many of the studies were quantitative and how may employed mixed methods.

You repeatedly stated 'we included 92 cross sectional studies and 3 cohort studies but these information (study designs) were not captured in any of the tables. Please include the study designs in Table 1.

How many studies satisfied the inclusion criteria and were included. It looks like table 4b is reporting on 116 studies instead of the 98 you previously mentioned in the abstract and the method section.

What is the "all" in table 4a and 4b indicating? please provide an interpretation in the results section.

Limitation: Your search for articles did not prioritize or consider longitudinal studies. You can therefore not make a categorical statement that "there was lack of longitudinal studies". You should rather state the reality which is your study did not consider longitudinal studies.

General comments: Some sentences were too long. it makes reading not interesting, causes reader fatigue and tend to loose its meaning at the tail end. otherwise it is a good publishable study.

6. PLOS authors have the option to publish the peer review history of their article (what does this mean?). If published, this will include your full peer review and any attached files.

**Do you want your identity to be public for this peer review?** For information about this choice, including consent withdrawal, please see our Privacy Policy.

Reviewer #1: No

Reviewer #2: No

Reviewer #3: No

Reviewer #4: No

Reviewer #5: No

Reviewer #6: No

Reviewer #7: **Yes: **Dorothy Serwaa Boakye

---

## [Decision Letter · Decision Letter 1]

9 Nov 2022

Alcohol use and associated risk factors among female sex workers in low- and middle-income countries: A systematic review and meta-analysis

PGPH-D-22-01100R1

Dear Dr. Beksinska,

We are pleased to inform you that your manuscript 'Alcohol use and associated risk factors among female sex workers in low- and middle-income countries: A systematic review and meta-analysis' has been provisionally accepted for publication in PLOS Global Public Health.

Best regards,

Palash Chandra Banik, MPhil

Academic Editor

Reviewer Comments (if any, and for reference):

Reviewer's Responses to Questions

**Comments to the Author**

1. If the authors have adequately addressed your comments raised in a previous round of review and you feel that this manuscript is now acceptable for publication, you may indicate that here to bypass the “Comments to the Author” section, enter your conflict of interest statement in the “Confidential to Editor” section, and submit your "Accept" recommendation.

Reviewer #2: All comments have been addressed

Reviewer #3: (No Response)

Reviewer #7: (No Response)

2. Does this manuscript meet PLOS Global Public Health’s publication criteria? Is the manuscript technically sound, and do the data support the conclusions? The manuscript must describe methodologically and ethically rigorous research with conclusions that are appropriately drawn based on the data presented.

Reviewer #2: Yes

Reviewer #3: (No Response)

Reviewer #7: Yes

3. Has the statistical analysis been performed appropriately and rigorously?

Reviewer #2: Yes

Reviewer #3: (No Response)

Reviewer #7: Yes

4. Have the authors made all data underlying the findings in their manuscript fully available (please refer to the Data Availability Statement at the start of the manuscript PDF file)?

Reviewer #2: Yes

Reviewer #3: (No Response)

Reviewer #7: Yes

5. Is the manuscript presented in an intelligible fashion and written in standard English?

Reviewer #2: Yes

Reviewer #3: (No Response)

Reviewer #7: Yes

6. Review Comments to the Author

Reviewer #2: (No Response)

Reviewer #3: (No Response)

Reviewer #7: The authors have addressed the major comments however, their response on how the scoring was done is still unclear and makes quality assessment of the included studies questionable. Specifically, it is still not clear how the scoring was done and how you arrived at the overall score of each study. The authors only stated '' scoring was based on cut offs from a previous systematic review by Beattie et al., (2020). Unfortunately, the authors, Beattie et al did not provide a clear-cut description of how their scoring was done. However, a review of their supplementary file 'S3 Text Quality assessment of quantitative data seem to show that Yes was scored as 1, No scored as 0 and a Yes to Q4(selection bias) and Q11 (Confounding factors) were scored as 0 (I'm only assuming). To be on a safer side, kindly email the corresponding author and find out how their scoring was done and how they arrived at the overall score for each study. Again, since you asserted, your scoring was based on Beattie et al study, one would expect that a study included in their systematic review which you also used would have the same score. For instance, i found that this study (Bitty - Anderson et al., 2019) was included in Beattie et al study and your current study, however, while Beattie et al score this study (Bitty-Anderson et al., 2019) 6, you scored that same study 9. Please carefully revise the scoring, quality assessment and make the necessary changes.

As part of your revision, you stated in the Methods section, Quality assessment, Line 234 that studies scoring weak quality were not included in the meta-analyses. However, these two studies; Duncan et al., 2010 and Munoz, 2005 scored low quality yet they were included in your meta- analyses (Check Fig 7). Please carefully review and revise accordingly

In your response to one of the comments of reviewer number 7 (Your search for articles did not prioritize or consider longitudinal studies. You can therefore not make a categorical statement that "there was lack of longitudinal studies".

You should rather state the reality which is your study did not consider longitudinal studies), You asserted that cohort studies are a type of Longitudinal studies. That is true but cohort studies can also be retrospective and not all longitudinal studies are cohorts. Other school of thoughts may still classify cohorts as prospective, longitudinal and retrospective. it's always safer to be specific in your reportage so you do not leave room for readers' assumptions. Please state the type of cohort study.

on a lighter note: I believe it would be better if you utilize ''et al'' since almost all the studies featured in this systematic review are co-authored.

7. PLOS authors have the option to publish the peer review history of their article (what does this mean?). If published, this will include your full peer review and any attached files.

**Do you want your identity to be public for this peer review?** For information about this choice, including consent withdrawal, please see our Privacy Policy.

Reviewer #2: No

Reviewer #3: No

Reviewer #7: **Yes: **Dorothy Serwaa Boakye
